# Molecular basis for the assembly of the Vps5-Vps17 SNX-BAR proteins with Retromer

Kai-En Chen [1], Vikas A. Tillu[1], Navin Gopaldass [2], Sudeshna Roy Chowdhury[2], Natalya Leneva[3], Oleksiy Kovtun [3], Juanfang Ruan [4], Qian Guo [1], Nicholas Ariotti [1], Andreas Mayer [2] & Brett M. Collins [1]✉

Retromer mediates endosomal retrieval of transmembrane proteins in all eukaryotes and was first discovered in yeast in complex with the Vps5 and Vps17 sorting nexins (SNXs). Cryoelectron tomography (cryoET) studies of Retromer–Vps5 revealed a pseudo-helical coat on membrane tubules where dimers of the Vps26 subunit bind Vps5 membrane-proximal domains. However, the Vps29 subunit is also required for Vps5–Vps17 association despite being far from the membrane. Here, we show that Vps5 binds both Vps29 and Vps35 subunits through its unstructured N-terminal domain. A Pro-Leu (PL) motif in Vps5 binds Vps29 and is required for association with Retromer on membrane tubules in vitro, and for the proper recycling of the Vps10 cargo in *Saccharomyces cerevisiae*. CryoET of Retromer tubules with Vps5–Vps17 heterodimers show a similar architecture to the coat with Vps5–Vps5 homodimers, however, the spatial relationship between Retromer units is highly restricted, likely due to more limited orientations for docking. These results provide mechanistic insights into how Retromer and SNX-BAR association has evolved across species.

Early endosomes are membrane bound cellular compartments that are essential for the sorting and maintenance of transmembrane cargo proteins and lipids. Transmembrane cargoes are delivered to the endosome from the cell surface or from intracellular organelles such as the Golgi and their trafficking is then controlled by a range of different peripheral membrane protein machineries. For the lysosomal degradative pathway, the cargoes are typically sorted by the Endosomal Sorting Complex Required for Transport machinery, while recycling of cargoes to either the plasma membrane or the trans-Golgi network is primarily driven by tubulovesicular carriers. The evolutionarily conserved Retromer complex is one of the key protein machineries controlling the cargo recycling and sorting at the endosome and is conserved from humans to the simplest single-celled eukaryotes[1–8]. Mutations in or altered expression of Retromer disrupt cellular homeostasis and have been strongly linked to a number of human

diseases including Alzheimer's[9–12], Parkinson's disease[13–16] and more recently carcinogenesis[17].

Retromer is composed of the three vacuolar protein sorting (Vps) subunits Vps26, Vps29 and Vps35. It was first identified in the budding yeast *Saccharomyces cerevisiae* as a heteropentameric complex with Vps5 and Vps17 responsible for the retrieval of cargoes such as the carboxy peptidase Y (CPY) receptor Vps10 from endosomes to the Golgi through the formation of tubulovesicular structures[18–20]. The core Retromer trimer acts as a scaffold for sorting cargoes, and functions with adaptors from the sorting nexin (SNX) protein family, including Vps5 and Vps17 in yeast, which control membrane association and cargo recruitment. Vps5 and Vps17 possess a phox homology (PX) domain that defines the SNX family and typically associates with membrane phosphoinositide lipids. This is preceded by an unstructured N-terminal domain of more than 100 residues in length and a

---

[1]Institute for Molecular Bioscience, the University of Queensland, St Lucia, QLD, Australia. [2]Department of Immunobiology, University of Lausanne, Epalinges, Switzerland. [3]Research Group Molecular Mechanism of Membrane Trafficking, Max Planck Institute for Multidisciplinary Sciences, Göttingen, Germany. [4]Electron Microscope Unit, Mark Wainwright Analytical Centre, University of New South Wales, Sydney, NSW, Australia. ✉e-mail: b.collins@imb.uq.edu.au

C-terminal bin/amphiphysin/rvs (BAR) domain that mediates homo and heterodimerization, oligomerisation and membrane tubulation. Although Retromer is conserved in higher metazoan species, the proposed SNX-BAR heterodimeric orthologues of SNX1/SNX2-SNX5/SNX6 are not stably associated with Retromer and can form a functionally distinct endosomal sorting complex (ESCPE-1)[21–25]. The human SNX-BAR proteins largely colocalise in with Retromer in endosomal retrieval domains, because they do not associate strongly with Retromer, it remains under debate whether they interact directly in human cells as compared to their yeast counterparts[2,26–30]. An additional difference is that the SNX5/6 subunits of ESCPE-1 can specifically recognize transmembrane cargo proteins via an unusual PX domain, including the cation-independent mannose-6-phosphate receptor and neuropilin-1[24,25,31]. In contrast, the Vps5-Vps17 proteins lack the unique PX domain structure and have so far not been identified to bind known cargo molecules. These studies highlight both similarities but also key differences between the Retromer and SNX-BAR machineries of higher and lower eukaryotes.

Structural studies are beginning to reveal some of the essential mechanistic features of Retromer assembly into a functional membrane coat. The cryoelectron tomography (CryoET) structure of Retromer with a homodimer of Vps5 from the thermophilic yeast *Chaetomium thermophilum* revealed the formation of membrane tubules whereby the PX and BAR domains of Vps5 were aligned to the membrane surface, and arches of Retromer formed oligomeric structures atop the Vps5 layer[32]. Subsequent structures of tubules coated by Retromer with both the yeast and metazoan Snx3 adaptor protein revealed a mechanism whereby the Retromer Vps26 subunits associate with the adaptor to form a convex scaffold contacting the membrane thus substituting the SNX-BAR in this role while preserving the overall complex architecture. In both Snx3 and SNX-BAR complexes, Vps29 resides at arch apexes distal from the adaptors and the membrane surface[32,33]. Despite the insights provided by these cryoET snapshots of Retromer there remains an important discrepancy between the observed structural organisation and studies demonstrating a key role of Vps29 in SNX-BAR association in vivo. Studies of these proteins in *S. cerevisiae* have shown that the interaction of Retromer with the Vps5–Vps17 proteins requires the presence of Vps29[18,34,35], which in cryoET structures is far from contacting the membrane associated PX and BAR domains[32,33]. In contrast, Vps26 is dispensable for the interaction in co-immunoprecipitations[35] although it is the primary subunit associated with Vps5 once assembled on membranes[32,33]. The N-terminal intrinsically disordered domain of Vps5, which is not visible in cryoET reconstructions, is also required for interaction with Retromer[34]. These results thus raise the question of how to reconcile the contrasting structural and in vivo data.

In this work we have resolved this controversy, using, biochemistry, crystallography, cryoEM, cell biology and in vitro membrane reconstitution experiments to define the precise mechanism by which Retromer forms a high affinity complex with Vps5 (and the associated SNX-BAR protein Vps17) required for membrane recruitment, membrane tubule formation, and trafficking. Combining Alphafold predictions and crystallography we found that tight association between Vps5 and Retromer depends on two factors: the interaction of Vps29 with Vps5 via a Pro-Leu (PL) motif and binding of the [K/R] Φ motif (Φ = hydrophobic residues) to the Vps29 and Vps35 interface. These sequences are unique to Vps5 homologues and are required for recruitment of Retromer by the functional Vps5–Vps17 heterodimer to form membrane tubule coats in vitro and for trafficking of vacuolar cargos in cells. This presents a mechanism of assembly of Retromer with SNX-BAR proteins first requiring a specific sequence in the disordered N-terminus of Vps5 for high affinity binding to the Vps35 and Vps29 subunits, which allows recruitment prior to the subsequent membrane-driven reorganization of the Retromer and SNX-BAR pentameric complex into the distinctive oligomeric coat.

## Results

### The N-terminal unstructured region of Vps5 mediates high affinity interaction with Retromer

Vps5 from the thermophilic yeast *Chaetomium thermophilum* (ctVps5) consists of PX and BAR domains preceded by an intrinsically disordered region (IDR) of ~150 residues (Fig. 1A, B). ctVps5 can form a homodimer via its BAR domain that associates with the trimeric ctRetromer complex to form membrane tubules, even in the absence of the heterodimeric partner ctVps17[32]. We revisited this interaction using GST pull-down assays involving the purified ctVps5, ctRetromer and the isolated domains/subunits of the proteins (Supplementary Fig. 1A–C). Full-length GST-ctVps5 binds strongly to both the ctRetromer trimer and the ctVps29–ctVps35 subcomplex in the absence of ctVps26 (Fig. 1C), which is in agreement with previously published results that Vps26 is dispensable for binding in solution[35]. Further examination using various truncation constructs of GST-ctVps5 showed that the ctVps29–ctVps35 subcomplex did not significantly interact with the PX and BAR domains but bound robustly to the N-terminal IDR of ctVps5 (residues 1–150) (Fig. 1C, D). Residues 51–100 of the ctVps5 IDR were required and sufficient for this interaction (Fig. 1D). We then carried out isothermal titration calorimetry (ITC) analysis to measure the binding affinity between ctRetromer and various truncation constructs of ctVps5. Consistent with our GST pull-down experiments, ctRetromer bound to ctVps5$_{51–100}$ with a $K_d$ of 660 nM (Fig. 1E and Table 1) whereas other Vps5 fragments including the PX and BAR domains failed to interact with Retromer in solution.

According to the structure-based sequence alignment of Vps5, the N-terminal IDR is much more divergent across species than the PX and BAR domains (Fig. 1F and Supplementary Fig. 2A). Interestingly, in the *Pezizomycotina* subdivision of fungi to which *C. thermophlilum* belongs, we identified a conserved dual PL-containing sequence motif ($^{73}$DPLGPL$^{78}$ in ctVps5), within the N-terminal IDR region that binds Retromer (Fig. 1F and Supplementary Fig. 2A). Notably, this sequence is reminiscent of the PL motifs previously shown to bind human Retromer subunit Vps29 such as in TBC1D5[36–38]. It is also identical in sequence to a macrocyclic peptide RT-D3 that we identified in previous in vitro screens for Retromer-interacting molecules using an approach called RaPID (random nonstandard peptide integrated discovery)[39]. The RaPID screen used human Retromer as bait and identified a series of macrocyclic peptides including RT-D3 that contain PL sequences that bind with high affinity and inhibit the conserved ligand binding site on Vps29. Vps5 sequences from other subdivisions including *Saccharomycotina* also contain a conserved DPL motif (Fig. 1F and Supplementary Fig. 2B, C), whereas presumed orthologs in higher eukaryotic species including human SNX1 and SNX2 do not possess conserved PL motifs in their unstructured N-terminal regions (Fig. 1F and Supplementary Fig. 3). Consistent with this, while ctRetromer bound the N-terminal domain of ctVps5 with a $K_d$ of 720 nM in ITC experiments, human SNX1 and human Retromer showed no detectable association under the same conditions (Fig. 1G).

### The PL-containing motif of Vps5 is required for Retromer association

The conservation of a PL-containing motif within the N-terminal IDR of yeast Vps5 led us to test if this sequence was essential for the association with Retromer. Using ITC, we found that the affinity of full-length ctVps5 and ctRetromer was reduced by the addition of the inhibitory cyclic peptide RT-D3[39] (Fig. 2A, B, Table 1, and Supplementary Fig. 4A, B). A mutant control peptide RT-D1 L7E that does not bind Vps29[39] had no effect on this association (Figs. 2A, B, Table 1 and Supplementary Fig. 4B). ctVps29 alone could bind ctVps5$_{51–100}$, but with a lower affinity ($K_d$ of 1.6 µM) compared to the full ctRetromer complex ($K_d$ of 660 nM; Fig. 2A and Table 1). In this instance, addition of the RT-D3 peptide completely blocks the binding. We also examined complex formation between ctRetromer and ctVps5 using mass

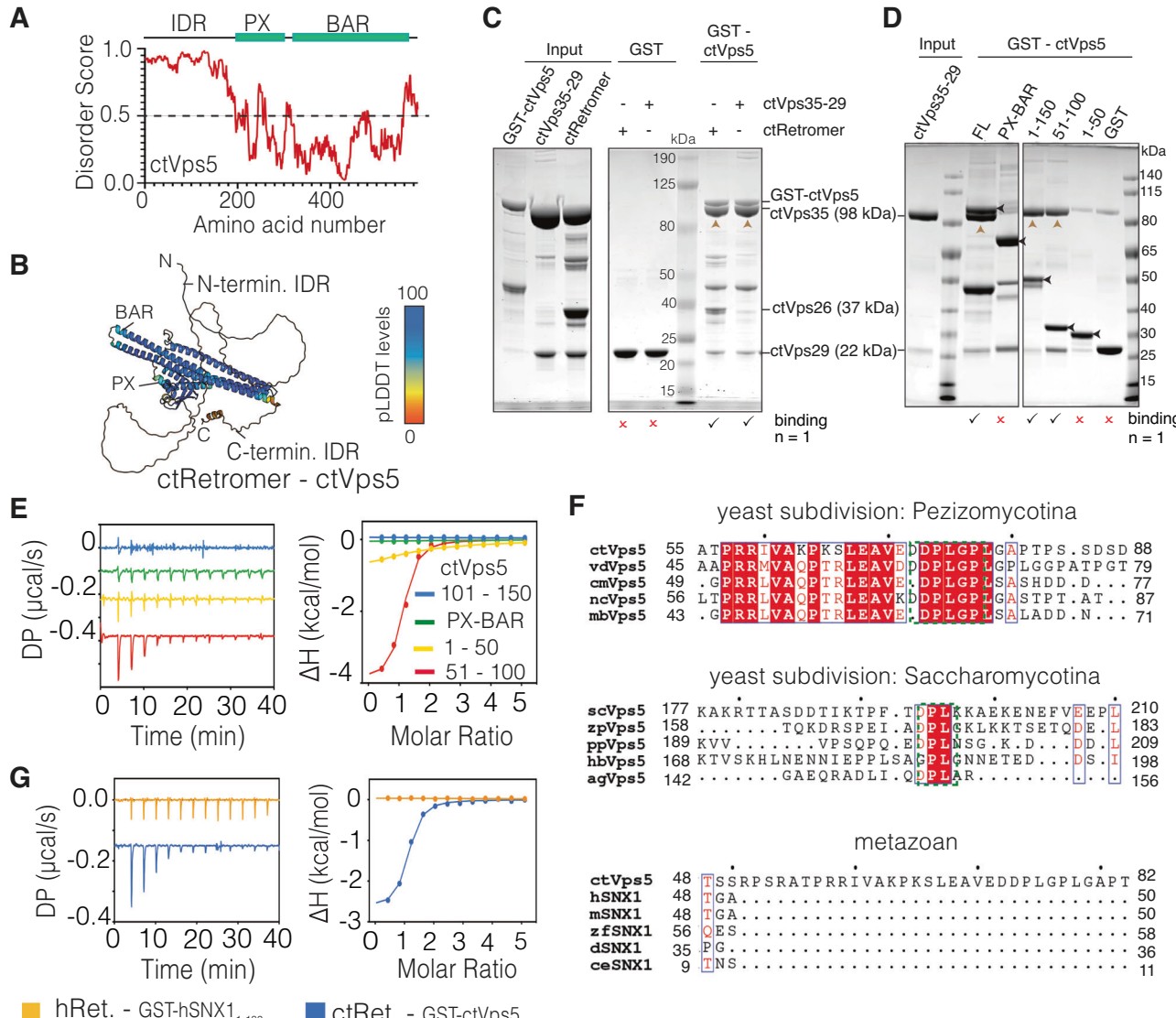

**Fig. 1 | Vps5 interacts directly with Retromer via its unstructured N-terminus.**
**A** Prediction of disordered sequences in ctVps5. Predicted regions of disorder were mapped using the D2P2 web server[87]. **B** Predicted structure of ctVps5 from the Alphafold2 database[88] coloured according to the per-residue confidence (pLDDT) score. The N-terminal IDR shows low confidence metrics (represented as green and orange) suggesting a lack of secondary or tertiary structure. **C** GST pull-down demonstrating the direct interactions between full-length GST-tagged ctVps5 and either the full ctRetromer complex or the ctVps29–ctVps35 subcomplex. Note, the lower band at ~47 kDa corresponds to a truncated GST-ctVps5 fragment that is present after affinity purification. **D** GST pull-down showing GST-ctVps5 unstructured N-terminal IDR (residues 1–150) is required and sufficient to bind the ctVps29–ctVps35 subcomplex. Black arrowheads indicate bands of full-length GST-ctVps5 constructs. For clarity, brown arrowheads indicate the ctVps35 bands

captured in the pull-downs. **E** ITC measurements between ctRetromer and various truncation constructs of ctVps5. **F** Sequence alignment of key regions of the Vps5 unstructured IDR showing the conserved PL motif (highlighted in red square) across different yeast species but not in metazoan ortholog SNX1. *ct, Chaetomium thermophilum; vd, Verticillium dahlia; cm, Cordyceps militaris; nc, Neurospora crassa; mb, Metarhizium brunneum; Sc, Saccharomyces cerevisiae; zp, Zygosaccharomyces parabailii; pp, Pichia_Pastoris; hb, Hyphophicia burtonii; Ag, Ashbya gossypii; h, Homo sapiens; m, Mus musculus; r, Rattus norvegicus; d, Drosophila melanogaster; z, Danio rerio; ce, Caenorhabditis elegans.* Extended sequence alignments are shown in Supplementary Fig. 1. **G** ITC measurements of ctRetromer and GST-ctVps5$_{1-185}$. Under the same conditions there is no detectable interaction observed between hRetromer and GST-hSNX1$_{1-139}$. All ITC graphs show the integrated and normalized data fit with a 1 to 1 binding ratio.

photometry (Fig. 2C). On its own ctVps5 in physiological salt conditions (200 mM NaCl) has a mass consistent with homodimer formation (via the C-terminal BAR domain), while ctRetromer appears primarily as a 1:1:1 heterotrimer under the same conditions. When ctRetromer–ctVps5 are combined in a 1:1 ratio we see formation of a complex with a mass consistent with one copy of ctRetromer bound to the ctVps5 dimer. Decreasing the ionic strength (40 mM NaCl) resulted in an increased mass corresponding to two copies of ctRetromer bound to the ctVps5 dimer consistent with previous studies showing ctRetromer can form a dimer of trimers (Supplementary Fig. 5A, B). In both cases, the complex was lost upon the addition of the cyclic

peptide RT-D3 (Fig. 2C), indicating that the ctRetromer and ctVps5 complexes no longer associate due to competition with the PL motif binding groove.

Next, we examined the membrane recruitment of ctRetromer by ctVps5 using liposome-binding experiments, measuring the fraction of proteins that are either co-pelleted together or remain in the supernatant. To test the importance of the PL-motif interaction we used the competitive cyclic peptide RT-D3 which blocks the binding site on Vps29, or the control RT-L4 which binds to the interface of Vps35–Vps26 and is not expected to affect ctVps5 interaction[39] (Fig. 2B, D, E). In control experiments, ctVps5 itself bound to

**Table 1 | Thermodynamic parameters for the binding of Retromer and Vps5/SNX1 by ITC**

| | $K_d$ (µM) | ΔH (kcal/mol) | ΔG (kcal/mol) | −TΔS (kcal/mol) |
|---|---|---|---|---|
| **ctRetromer** | | | | |
| ctVps5 | 0.72 ± 0.01 | −8.50 ± 0.94 | −8.39 ± 0.01 | −0.35 ± 1.18 |
| GST-ctVps5$_{1–185}$ | 0.72 ± 0.01 | −3.10 ± 0.60 | −8.38 ± 0.01 | −5.29 ± 0.62 |
| GST-ctVps5$_{1–50}$ | 40.07 ± 8.56 | −2.70 ± 0.34 | −6.00 ± 0.13 | −3.31 ± 0.47 |
| GST-ctVps5$_{51–70}$ | No binding detected | | | |
| GST-ctVps5$_{61–80}$ | 60.07 ± 12.24 | −7.22 ± 1.94 | −5.76 ± 0.12 | 1.45 ± 2.07 |
| GST-ctVps5$_{51–100}$ | 0.66 ± 0.01 | −4.10 ± 0.6 | −8.44 ± 0.01 | −4.31 ± 0.54 |
| GST-ctVps5$_{51–100}$ PL74AA | 5.04 ± 0.27 | −1.76 ± 0.23 | −7.25 ± 0.07 | −5.49 ± 0.21 |
| GST-ctVps5$_{51–100}$ RKL/AAG | No binding detected | | | |
| GST-ctVps5$_{101–150}$ | No binding detected | | | |
| ctVps5$_{PX-BAR}$ | No binding detected | | | |
| ctVps5$_{53–80}$ peptide | 0.74 ± 0.05 | −2.83 ± 0.54 | −8.38 ± 0.03 | −5.57 ± 0.60 |
| ctVsp5$_{71–80}$ peptide | No binding detected | | | |
| GST-ctVps17$_{1–163}$ | No binding detected | | | |
| **ctRetromer + RT-D3** | | | | |
| ctVps5 | 9.41 ± 0.75 | −10.48 ± 0.85 | −6.86 ± 0.05 | 4.25 ± 0.39 |
| **ctRetromer + RT-D1 L7E** | | | | |
| ctVps5 | 0.75 ± 0.05 | −7.53 ± 0.10 | −8.37 ± 0.03 | −0.81 ± 0.09 |
| **ctVps29** | | | | |
| GST-ctVps5$_{51–100}$ | 1.60 ± 0.03 | −2.16 ± 0.60 | −7.87 ± 0.06 | −5.75 ± 0.61 |
| RT-D3 | 0.59 ± 0.02 | −7.56 ± 0.04 | −8.40 ± 0.09 | −0.93 ± 0.03 |
| RT-D1 L7E | No binding detected | | | |
| **hRetromer** | | | | |
| GST-mSNX1$_{1–139}$ | No binding detected | | | |
| **mVps29** | | | | |
| GST-ctVps5$_{1–185}$ | 4.82 ± 0.95 | −0.91 ± 0.36 | −7.20 ± 0.05 | −6.17 ± 0.51 |
| ctVsp5$_{71–80}$ peptide | 54.10 ± 7.7 | −8.00 ± 1.10 | −5.80 ± 0.10 | −2.10 ± 0.90 |

membranes composed of Folch I lipids supplemented with the phosphoinositide PtdIns3*P* which binds the ctVps5 PX domain, and this was not affected by either peptide. The ctRetromer complex was robustly recruited to PtdIns3*P*-containing liposomes in the presence of ctVps5 in agreement with previous results (Fig. 2D, E and Supplementary Fig. 6A)[32]. The cyclic peptide RT-L4 did not affect liposome recruitment of ctRetromer by ctVps5 as expected, whereas RT-D3 significantly reduced the amount of ctRetromer in the pelleted fraction with ctVps5 (Fig. 2D, E). We still observed the formation of membrane tubules by ctVps5 in the presence of RT-D3, suggesting that blocking the PL motif interaction does not prevent the BAR-domain protein from forming tubules (Supplementary Fig. 6B). In agreement with the solution binding studies, these results show that interaction of the ctVps5 PL motif with Vps29 is important for ctVps5-mediated membrane recruitment of the ctRetromer complex.

## Structural basis for Vps5 interaction with Retromer

To better understand how Vps5 and Retromer interact at the molecular level we first used Alphafold2-multimer[40,41] as implemented in ColabFold[42] to model the potential binding interfaces. The predicted alignment error (PAE) plots and interfacial pTM (iPTM) scores showed a strong correlation between ctRetromer and a highly specific N-terminal region of ctVps5 (Fig. 3A and Supplementary Fig. 7A). Across multiple models, we identified a high-confidence binding sequence within the N-terminal IDR of ctVps5 that matched precisely with the region containing the PL motif identified in our biochemical experiments (Fig. 3B, C and Supplementary Fig. 7B). In all predicted models, this sequence adopts an extended conformation, with N-terminal residues forming direct contacts with both ctVps35 and ctVps29 near the interface of the two proteins, with the key [73]DPLG[76] residues forming a short turn-conformation that docks into the

conserved hydrophobic pocket of ctVps29 (Fig. 3B, C). But unlike the PL motif binding pocket, the ctVps35 and ctVps29 interface responsible for binding to the N-terminal residues of ctVps5 is rather diverse across different yeast species. This suggests that the PL motif may act as the primary conserved binding region, while the ctVps35 and ctVps29 interface is likely to play a more supportive or specificity-defining role.

To test the prediction from Alphafold2, we next determined the structure of Vps29 in complex with a ctVps5 peptide containing the core [73]DPL[75] motif. Initially, a peptide composed of residues [53]SRATPRRIVAKPKSLEAVEDDPLGPLGA[80] containing both the predicted Vps35 and Vps29-binding sequences was screened. Its binding affinity for ctRetromer ($K_d$ of 740 nM) corresponded well to that of the full-length ctVps5 ($K_d$ of 720 nM) (Fig. 3D and Table 1). Attempts to obtain crystals of ctRetromer bound to the ctVps5$_{53–80}$ peptide were unsuccessful. However, after screening a variety of constructs and species, crystals of mouse VPS29 (mVPS29) in complex with a shorter version of the ctVps5 peptide [72]DDPLGPLGA[80] were obtained (Fig. 3E and Supplementary Fig. 7C). The peptide-bound crystal structure was solved at 1.7 Å revealing two molecules in the asymmetric unit (Fig. 3F, G and Supplementary Table 1). The ctVps5$_{71–80}$ peptide residues D72 to G79 were clearly delineated in the electron density map and confirm that it binds to the conserved hydrophobic groove of VPS29 consisting of Leu152, Tyr163 and Tyr165 (Leu159, Tyr170 and Tyr172 respectively in *C. thermophilum*). Overall, the structure reveals a similar binding arrangement to the previously published crystal structure of ctVps29−RT-D3 complex (Supplementary Fig. 8A–C). On closer inspection, the ctVps5 peptide adopts a slightly different registration in the crystal structure compared to the AlphaFold2 predicted models (Fig. 3H, I and Supplementary Fig. 8C). The sequence [73]DPLG[76] is invariably predicted by AlphaFold2 to form the primary

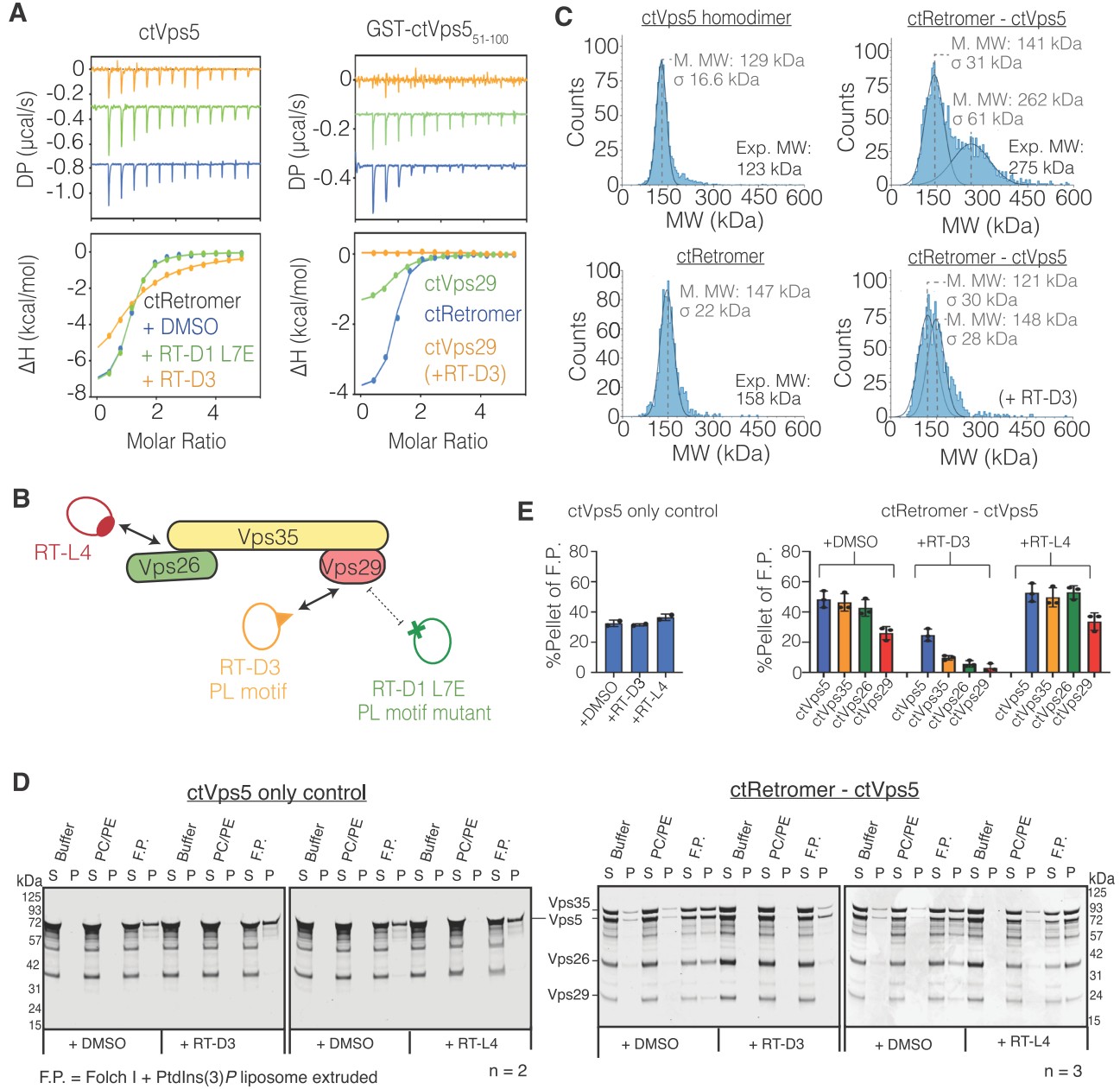

**Fig. 2 | The PL-containing motif of Vps5 is required for Retromer association.**
**A** ITC measurements of ctVps5 (left) and GST-ctVps5$_{51-100}$ (right) with ctRetromer or ctVps29 in the presence or absence of Vps29-binding cyclic peptides[39].
**B** Schematic diagram showing the cyclic peptides applied in the ITC and liposome pelleting assay. RT-D3 binds strongly to Vps29 and inhibits interactions of other PL motif-containing proteins. RT-D1 (L7E) is a mutant peptide that does not bind Vps29 and acts as a negative control. RT-L4 acts as a molecular staple that binds to the Vps26–Vps35 interface and does not affect Vps29 from binding to PL motif-containing proteins. **C** Mass photometry of ctVps5, ctRetromer and the ctVps5–ctRetromer complex in the presence and absence of RT-D3 inhibitor.

**D** Liposome-pelleting assays of ctVps5 and ctRetromer−ctVps5 complex in the presence of indicated cyclic peptides. Unilamellar vesicles were composed of either PC/PE control lipid, or Folch I lipids supplemented with PtdIns(3)P. "S" and "P" indicates unbound supernatant and bound pellet respectively. **E** Quantitation of Liposome-binding assay as shown in (**D**). The y-axis indicates the percentage of protein pelleted with F.P. liposome. Blue, yellow, green, and red indicates the ctVps5, ctVps35, ctVps26 and ctVps29 containing either DMSO, RT-D3 or RT-L4 cyclic peptides. Data represent the mean ± standard deviation of at least two independent experiments for the ctVps5 only control and three independent experiments for the ctRetromer−ctvps5 complex.

contact with the ctVps29 binding pocket residues Leu159, Tyr170 and Tyr172 (using *C. thermophilum* numbering), while in the mouse VPS29 crystal structure the primary contact in both molecules in the asymmetric unit is shifted to an adjacent PL motif formed by ctVps5 residues $^{76}$GPLG$^{79}$. In the crystal structure, the ctVps5 Asp73 side chain interacts with Lys173 in mVPS29, promoting a second contact between ctVps5 Asp72 and mVPS29 Lys21 (Fig. 3I, J). In ctVps29 the equivalent to mVPS29 Lys173 is Ala185 which in AlphaFold2 predicted models contacts Val70 in ctVps5. We therefore propose that the binding

orientation in the crystal structure (binding mode B) is likely reflecting a weaker secondary conformation stabilised by Lys173 in mVPS29, while that predicted by AlphaFold2 is the primary mode for yeast protein interactions (binding mode A) (Fig. 3J).

### High affinity interaction with Vps5 is due to avidity of binding both Vps29 and Vps35

According to our modelling, residues 53 to 71 of ctVps5 (upstream of the core PL motif) form an extended contact with the ctVps29−ctVps35

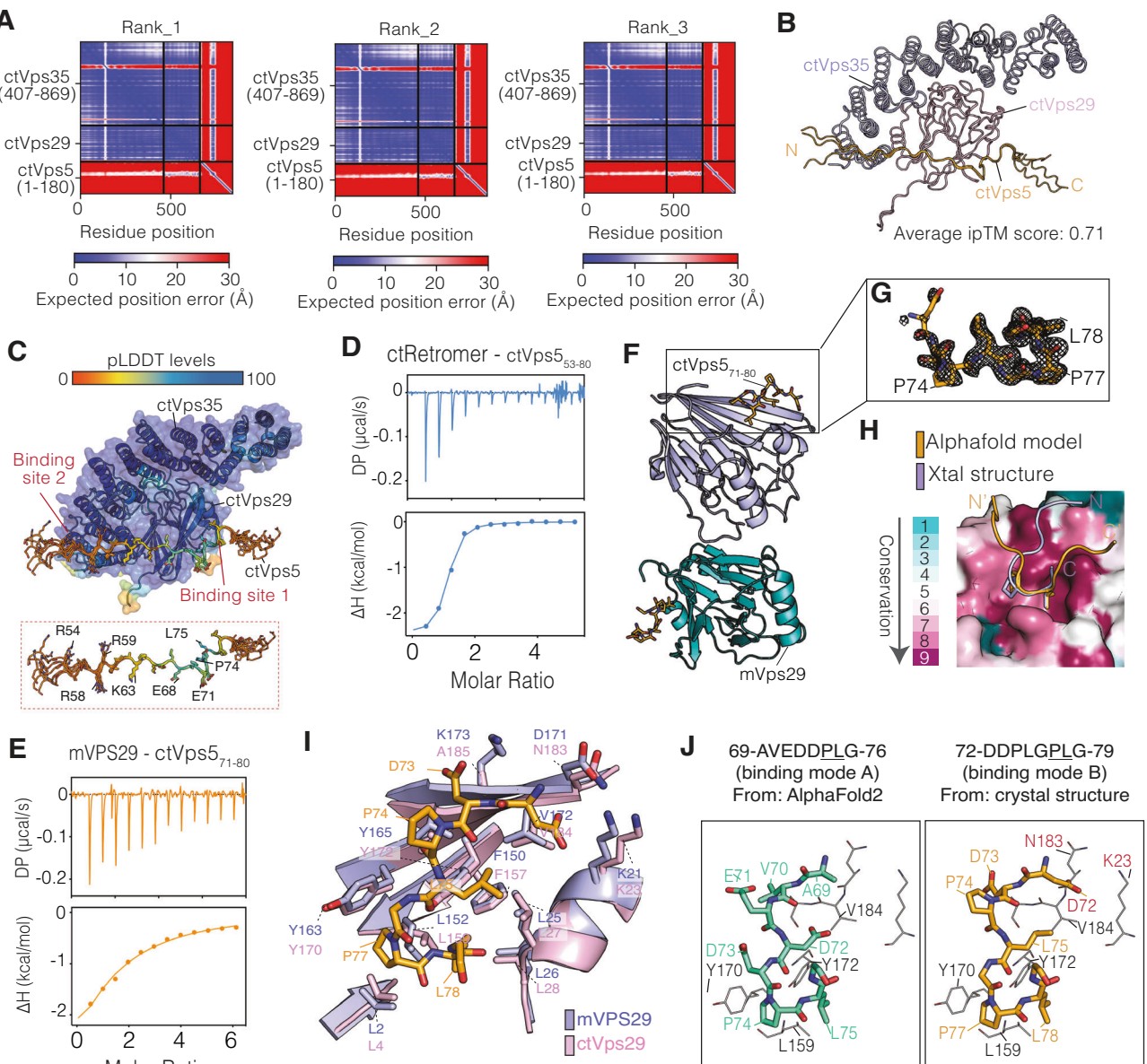

**Fig. 3 | Structural basis for Vps5 interaction with Retromer. A** PAE plots showing the three top-ranked Alphafold2 predicted models of the trimeric ctVps35$_{407-869}$–ctVps29–ctVps5$_{1-180}$ subcomplex. **B** The top-ranked model of ctVps35–ctVps29 is shown in ribbon diagram, with the bound region of ctVps5$_{1-180}$ of all three predictions overlaid. The close overlap of the three predicted binding regions of ctVps5$_{1-180}$ suggests a confidently predicted interaction. **C** As in (**B**) but with the predicted subcomplex model coloured according to the pLDDT score. The conserved PL motif binding pocket (site 1) and the ctVps29–ctVps35 interface (site 2) are indicated. The key ctVps5 residues involved in binding to the ctVps29–ctVps35 interface are shown in further detail in the expanded box. **D** ITC measurements showing binding of ctRetromer to a synthetic ctVps5$_{63-80}$ peptide. **E** ITC measurements showing binding of mVPS29 to a short synthetic ctVps5$_{71-80}$ peptide. **F** Crystal structure of the mVPS29–ctVps5$_{71-80}$ peptide complex showing

two molecules in the asymmetric unit. For clarity, the ctVps5$_{71-80}$ peptide is shown in stick mode. **G** Electron density of the ctVps5$_{71-80}$ peptide, corresponding to a simulated-annealing OMIT Fo-Fc map contoured at 3σ. **H** Sequence conservation mapped onto the mVPS29 protein surface highlights the PL motif binding surface. The ctVps5$_{71-80}$ peptide observed in the crystal structure is shown in light orange, and the Alphafold2 prediction of same sequence bound to ctVps29 are shown in stick representation for comparison. **I** Overlay of the mVPS29–ctVps5$_{71-80}$ crystal structure (light purple) with the previously determined apo ctVps29 crystal structure[32] (pink; PDB ID: 5W8M) showing the key residues involved in binding. **J** Structural models showing how ctVps5 binds to ctVps29 through its dual PL motif binding mechanism. Left and right panels represent the GPLG and DPLG binding mode respectively.

interface (Fig. 4A, B). This mechanism appears to be relatively conserved as AlphaFold2-derived models of the complex from various yeast species including *S. cerevisiae* and *S. pombe* invariably predict that Vps5 forms a similar extended interaction (Fig. 4A and Supplementary Fig. S9A–C). Unlike the conserved Vps5 PL motif however, the Vps5 residues that associate with the Vps29–Vps35 interface are relatively diverse across the different yeast Vps5 orthologues (Supplementary Fig. 2A–C). Interestingly, the extended binding motifs in these

three representative species contain multiple lysine or arginine side-chains followed by hydrophobic residues that we refer to as the [K/R]Φ motif (where Φ is any bulky hydrophobic sidechain) contacting residues at the Vps29 and Vps35 interface (Fig. 4B–D). Specifically, we found that Arg59 of ctVps5 (Arg152 and Arg168 of scVps5 or Lys59 of spVps5) forms electrostatic contacts with the conserved Glu763 of ctVps35 (Glu418 of scVps35 and Glu329 of scVps35) (Fig. 4B–D). Both the PL-motif and the [K/R]Φ sequences contribute to Retromer

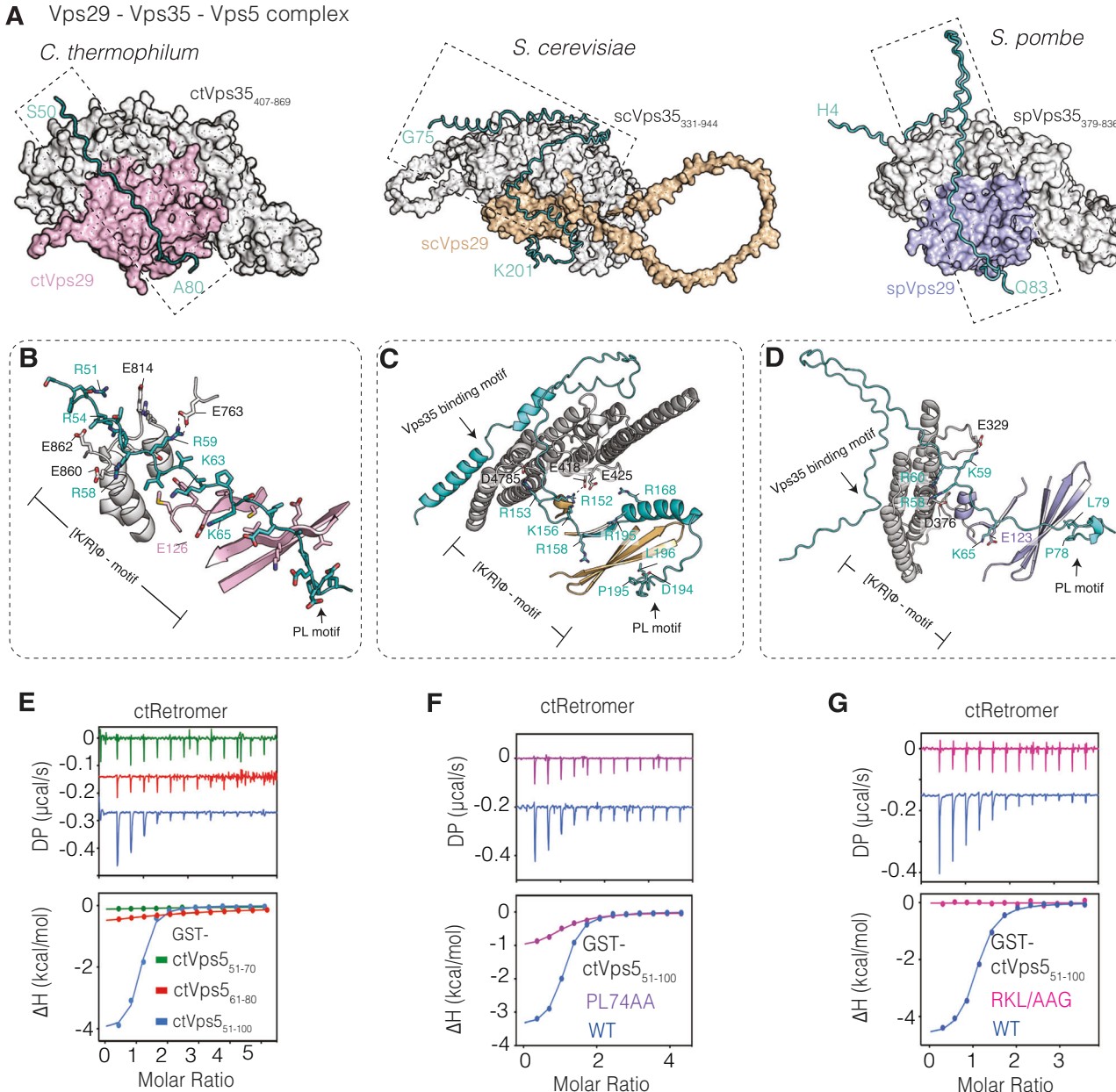

**Fig. 4 | High affinity interaction with Retromer involves multiple sequences within the Vps5 N-terminal domain. A** Surface representation of the Vps29–Vps35–Vps5 Alphafold2-derived models from *C. thermophilum*, *S. cerevisiae* and *S. pombe*. The key regions of Vps5 involved in binding are shown as $C_\alpha$ ribbons. **B** Cartoon representation highlighting the key Lys, Arg and hydrophobic residues upstream of ctVps5, (**C**) scVps5 and (**D**) spVps5 PL motifs involved in binding. **E** ITC measurements showing the binding between ctRetromer and various truncation constructs of GST-tagged ctVps5, (**F**) ctVps5 PL74AA mutant (P74A/L75A) and (**G**) ctVps5 RKL/AAG mutant (R50A/R53A/R58A/R59A/K63AK65A/L67G). Both PL74AA and RKL/AAG mutant loss their binding affinity to ctRetromer compared to the wild-type under the same conditions. All ITC graphs show the integrated and normalized data fit with a 1 to 1 binding ratio.

binding affinity, as the ctVps5$_{51–70}$ and ctVps5$_{61–80}$ constructs containing either the [K/R]Φ or the PL motifs, respectively, bind very weakly to ctRetromer when compared to ctVps5$_{51–100}$, which contains both motifs (Fig. 4E and Table 1). Similarly, the short ctVps5$_{71–80}$ peptide contains mainly the PL motif also reveals very weak binding to ctRetromer compares to the long ctVps5$_{53–80}$ peptide (Supplementary Fig. 10A). In the reciprocal experiment, a ctVps5$_{51–100}$ construct with the PL motif at position 74 and 75 substituted by alanine resulted in much weaker binding to ctRetromer, although it did not completely abolish the interaction (Fig. 4F). In contrast, the ctVps5$_{51–100}$ construct with the [K/R]Φ motif (R50A, R53A, R58A, R59A, K63A, K65A and L67G) mutated to alanine and glycine resulted a complete loss of binding (Fig. 4G). The Alphafold model and

biochemical studies thus show a role of both the [K/R]Φ and PL motifs in associating with Retromer.

In addition, we observed that AlphaFold2 models of *S. cerevisiae* and *S. pombe* Vps5 predicted a third extended peptide interaction with the rear-side of the Vps35 α-helical solenoid, which is not seen in the *C. thermophilum* predictions (Fig. 4A and Supplementary Fig. 10B, C). This interaction involves Leu and Phe sidechains of Vps5 binding the C-terminal end of Vps35, remarkably resembling the LFa repeat motifs (Leu, Phe and acidic sidechains) in the human FAM21 protein that, as we recently showed, bind to the same region of human VPS35[43] (Supplementary Fig. 10D). Using GST pull-down, we confirmed that hRetromer was able to bind to the GST-tagged scVps5$_{70–128}$ construct containing the LF motif (Supplementary Fig. 10E). In comparison to the

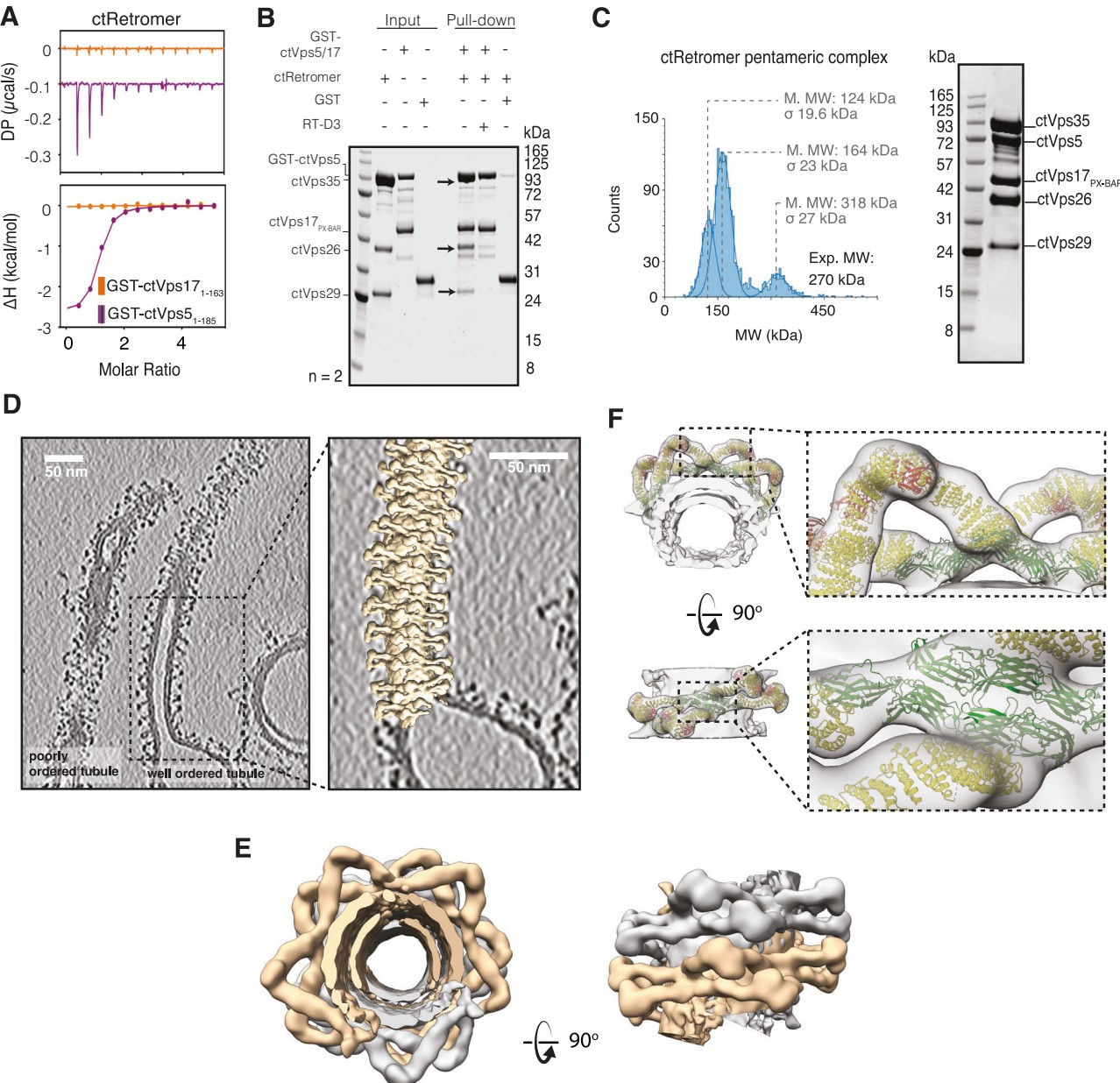

**Fig. 5 | Assembly of Retromer with the Vps5–Vps17 heterodimeric complex on membrane tubules. A** ITC measurements of ctRetromer with GST-ctVps17$_{1-163}$ or GST-ctVps5$_{1-185}$. Under the same conditions and concentrations no interaction is detected with the Vps17 N-terminal domain. **B** GST pull-down demonstrating the direct interaction between GST-ctVps5–ctVps17$_{PX-BAR}$ and ctRetromer. The interactions are blocked by adding inhibitory cyclic peptide RT-D3. **C** Mass photometry of heteropentameric complex composed of full-length ctVps35, ctVps26, ctVps26, ctVps5 and ctVps17$_{PX-BAR}$. The SDS-PAGE gel of the complex is shown on the right. **D** Left panel shows a slice from a tomographic reconstitution showing examples of coated tubules with well-ordered and poorly ordered Retromer complexes side by side. The right panel shows the overlay of the tubule with a model that was generated by placing copies of the cryoEM map at positions and orientations determined by STA. See also Supplementary Movies 2 and 3. **E** Close up of the tubule model shown in profile and cross-section. The coat is formed by the two chains of Retromer arches (separate helical rows are clored in grey and caramel for clarity). These form pseudo-helices winding around the membrane tubule. This arrangement lacks strict helical symmetry as the pitch and units per turn adapts to the tubule diameter. **F** Molecular model generated by rigid body fit of ctRetromer[39] to the cryoEM map. The specific arrangement of two adjacent Vps26 dimers (in green) was not seen in the previous Retromer structure using Vps5 alone[39]. Note that the PX-BAR domains of the Vps5-Vps17 heterodimers are not resolved, and seen only as a dense membrane-proximal layer.

GST-tagged hFAM21$_{R19-R21}$ construct, the LF motif alone only binds weakly to VPS35, we speculate that Vps5 orthologues containing these three sequences together will create a high binding affinity for Retromer due to enhanced avidity.

### Assembly of Retromer with the Vps5–Vps17 heterodimeric complex on membrane tubules

In yeast, Vps17 is a functional partner of Vps5, which form heterodimers via their C-terminal BAR domains[18,34,44,45]. Vps17 also has a long

N-terminal unstructured domain but does not have a conserved PL motif (Supplementary Fig. 11) and showed no predicted interactions with Retromer using AlphaFold2 (not shown). Consistent with this, by ITC the unstructured N-terminal region of ctVps17 (ctVps17$_{1-185}$) did not show any observable binding to ctRetromer compared to ctVps5$_{1-185}$ (Fig. 5A). In parallel experiments, pull-down of the GST-tagged ctVps5–ctVps17 heterodimer showed strong binding to ctRetromer which was lost upon the addition of RT-D3 (Fig. 5B). This suggests that the PL motif-containing N-terminal domain of ctVps5 is

primarily responsible for heteropentameric complex assembly in solution without any significant contribution from ctVps17.

The structure of ctRetromer on membrane tubules was previously revealed by cryoET, in complex with ctVps5 as a membrane-associated homodimer[32,33]. In this complex there are two copies of the ctVps5 N-terminal domain available for ctRetromer binding and recruitment, although the resolution of this structure did not allow visualisation of the ctVps5 N-terminus. We next sought to confirm that a similar membrane-assembled Retromer coat would be formed in the presence of the physiologically relevant Vps5–Vps17 heterodimer where only a single PL motif is available. To address this, we first generated the pentameric complex composed of core ctRetomer, full-length ctVps5 and ctVps17$_{PX-BAR}$ (PX and BAR domains of ctVps17) (Fig. 5C). The truncated ctVps17$_{PX-BAR}$ complex was used because it was more stable than the full-length ctVps17 protein following purification, and as shown above, the N-terminal region of Vps17 is not required for ctRetromer interaction (Fig. 5A, B). Negative stain electron microscopy of the ctRetromer–ctVps5–ctVp17 heteropentameric complex on Folch liposomes supplemented with PtdIns3*P* revealed extended membrane tubules (Supplementary Fig. 12A). These had a diameter of ~40 nm and at this resolution were indistinguishable from the tubules generated by the ctRetromer–ctVps5 complex (Supplementary Fig. 12B). In cryoEM and cryoET imaging, these tubules also showed a distinctive protein coat with an average diameter ~40 nm (compared to ~26.3 nm for the membrane layer) (Fig. 5D, Supplementary Fig. 12C, D and Supplementary Movie 1) similar in appearance to those of ctRetromer–ctVps5[32,33], with visible Retromer arches. The ratio of protein coated tubules over round shaped vesicles per image was ~1:1. We then collected a cryoET dataset and resolved the coat architecture by a subtomogram averaging (STA) in a subset of this data (Supplementary Table 2). We first picked all tubules for reference-free STA. The resulting cryoEM map poorly resolved the features in the z-direction of tomographic reconstructions. This is a common problem due to the lack of resolution in this direction in tomographic reconstructions and preferential orientation of Retromer arches in the coat. We circumvented this by focusing further STA on the most well-ordered tubules, which resulted in *a* ~ 32 Å cryoEM map with isotropic resolution (Fig. 5E and Supplementary Fig. 12E, F). Together with coordinates and orientations of subtomograms, this map allowed us to visualise the coat architecture, and model Retromer subunit arrangements by rigid body fitting. On tubules with well-ordered Retromer coats, the lattice is formed by two rows of arches that wind around the membrane tubules (Fig. 5D–F and Supplementary Movies 2 and 3). This forms a pseudo-helical arrangement, lacking strict helical symmetry as it accommodates various membrane diameters and gaps in structure, and is a known feature of the Retromer coat[32,33]. However, in this case the packing of the coat involves a close abutment of Vps26 dimers in adjacent rows which form the membrane proximal attachment point for the extended Retromer arches. Unfortunately, the resolution does not allow clear visualisation of the PX-BAR domains of Vps5 and Vps17 heterodimers, which are seen only as a dense protein layer on the membrane (Fig. 5E, F). However, the cryoET data suggests that compared to when the PX-BAR layer is composed of a Vps5 homodimer, where redundancy of docking points for Vps26 dimers results in multiple possible spatial arrangements between the dimers occurring in the same coat[32], the presence of the Vps5–Vps17 heterodimer presents a more restricted set of docking sites for the Vps26 dimers that results in a coat with a single predominant organisation.

## The PL motif is required for recruitment of Retromer to supported membrane tubules by *S. cerevisiae* Vps5

To examine the impact of the Vps5 N-terminus on the dynamics of pentameric coat formation we turned to a supported membrane tube (SMT) assay, which has been established for the complex from *S. cerevisiae*[44,46]. As described above, the scVps5 N-terminal domain is predicted to make similar but even more extensive contacts than ctVps5 with the Vps29 and Vps35 proteins. The *S. cerevisiae* pentameric complex was assembled from purified recombinant scRetromer carrying mRuby and scVps5–scVps17 heterodimer (scSNX-BAR) tagged with GFP. Incubation of SMTs with these purified proteins allowed us to observe coat growth in real time (Fig. 6). The impact of the scVps5 PL motif on growth of the coat was tested by addition of the competing peptide RT-D3, or of the control peptide RT-D3 with a scrambled sequence (Supplementary Fig. 4A). In control experiments, scSNX-BAR-GFP and scRetromer-mRuby were co-recruited to the tubes within a minute after their addition, and after 2 to 3 min strong fluorescence signals concentrated at multiple discrete sites, suggesting the formation of distinct oligomeric segments on the SMTs (Fig. 6A, B and Supplementary Movie 4). In the similar experiment where we applied the quadruple mutant Vps5 (L121G/F122G/F161E/L196R), which alters residues in the three main sites of predicted interaction–the PL motif, the [K/R]Φ motif, and the Vps35-binding LFa sequences, showed a defect of coat formation (Supplementary Fig. 13A, B and Supplementary Movies 7 and 8). The quadruple mutations on Vps5 do not impair heterodimer formation and able to form oligomer on the SMT (Supplementary Fig. 13C, D). The lipid tracer Cy5.5-PE showed a reciprocal loss of fluorescence at the sites of the oligomeric segments (Fig. 6B–E and Supplementary Fig. 13E). In our previous analyses of such effects in this system, we interpret this loss of Cy5.5-PE as a constriction of the membrane tubes by the coat[44]. While the scrambled peptide-D3 did not affect coat formation and membrane constriction, the RT-D3 peptide that competes with the PL-motif binding to Vps29 strongly reduced scRetromer recruitment and prevented membrane constriction (Fig. 6A–E and Supplementary Movie 5). This underlines the importance of the Vps5 N-terminus, and the PL motif in particular, for Retromer interaction.

As a further test of specificity of the cyclic peptide competitor we exploited the fact that at higher concentrations (100 nM), the scSNX-BAR complex alone suffices to form oligomeric clusters that constrict the SMTs[44]. Since the scSNX-BAR complex did not carry a fluorescent label we used the reduction of lipid fluorescence, which results from coat formation and tube constriction, as an assay. Under these conditions, the cyclic peptide RT-D3 had no effect on coat formation and membrane constriction, supporting the notion that its effect is due to blocking scRetromer association (Fig. 6F and Supplementary Movie 6). Together, these SMT assays confirm that the interaction between the Vps5 PL motif and scRetromer is a prerequisite for pentameric complex assembly and allows Retromer to promote coat formation and membrane constriction by the scSNX-BAR proteins.

## The interaction of Retromer with the Vps5 N-terminal domain is required for Vps10 recycling

Our data illustrates the importance of the Vps5 N-terminal unstructured domain and its conserved PL motif in pentameric complex assembly both in solution and on synthetic membranes in vitro. Lastly, we tested the relevance of these sequences for the Vps5-Retromer interaction using two assays: The first exploits the strong fragmentation of vacuoles that results from a loss of Vps5-Retromer association. In this case, the freed Retromer integrates into the endolysosomal membrane fission-promoting complex CROP[47]. The resulting excessive fission activity leads to a strong fragmentation of the vacuoles into clusters containing a high number of small vesicles, which can be visualized after staining the vacuolar membrane with the vital dye FM4–64. A second indicator of Vps5-Retromer interaction is the sorting of Vps10, a well-characterized CPY receptor cargo, which is recycled from endosomes to the Golgi using both Retromer and SNX-BARs[19,20]. We generated *S. cerevisiae VPS5* knockout cells (*vps5Δ*) expressing a C-terminal, genomic Vps10-mNeonGreen fusion. *vps5Δ* cells were rescued with *vps5* alleles carrying mutations that should block interactions of its N-terminal domain with Retromer (Fig. 7A).

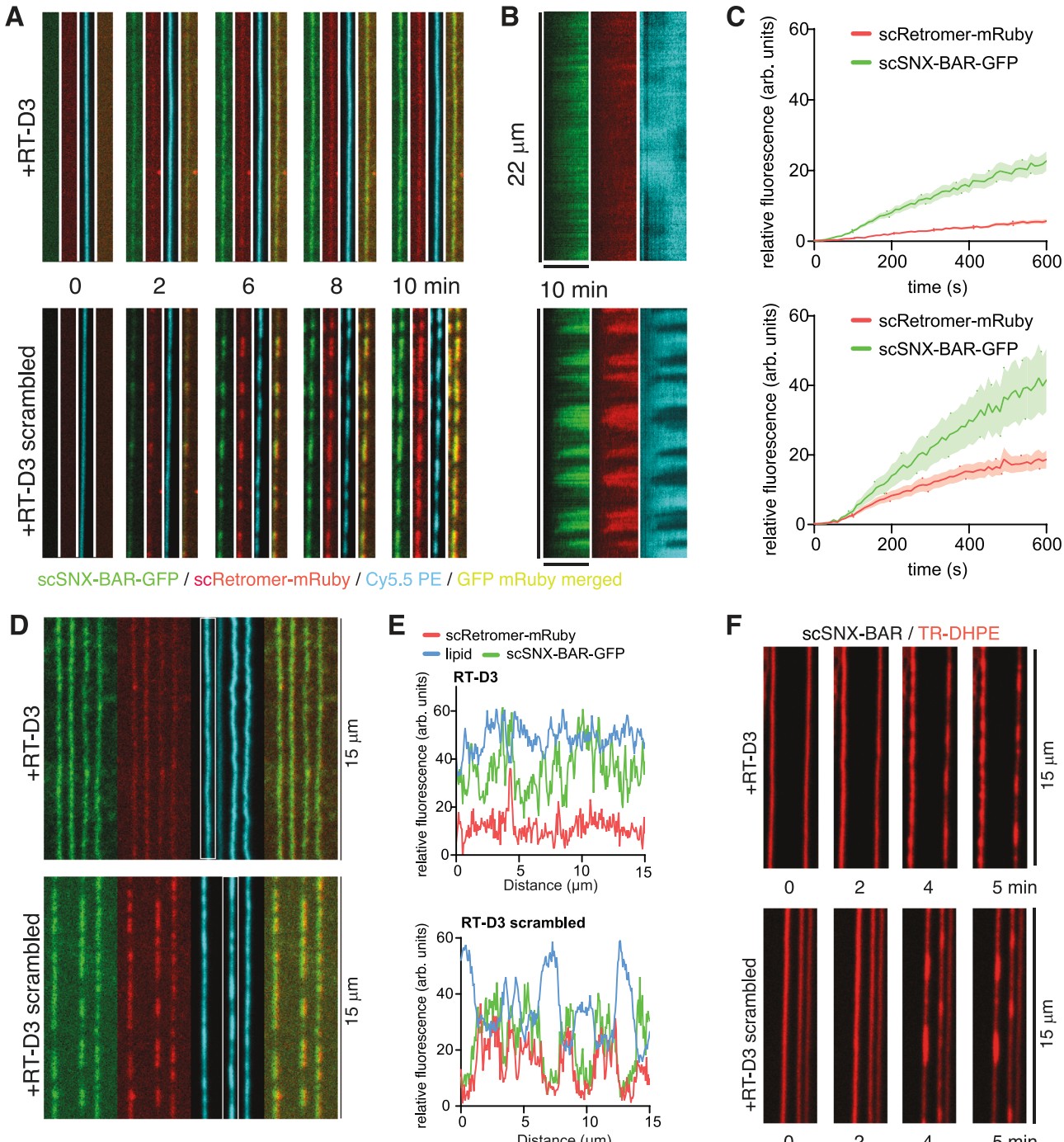

**Fig. 6 | The PL motif of *S. cerevisiae* Vps5 is required for recruitment of Retromer to supported membrane tubules. A** Effect of RT-D3 and RT-D3 scrambled cyclic peptides on scSNX-BAR-GFP and scRetromer-mRuby recruitment on supported membrane tubes (SMTs). SMTs containing 5% PtdIns(3)*P* were formed and incubated with 25 nM scSNX-BAR-GFP and 50 nM scRetromer-mRuby while being imaged by confocal microscopy. Frame rate: 0.1 Hz, cyclic peptide concentration: 300 μM. See also Supplementary Movies 4 and 5. Green: scSNX-BAR-GFP, red scRetromer-mRuby, blue: Cy5.5-PE. **B** Kymograph of SMTs shown in **A** over the time course of the whole experiment (10 min). **C** Quantification of the recruitment of scSNX-BAR-GFP and scRetromer-mRuby on the tubule. Quantification was done using FIJI, measuring the mean fluorescence signal on the tubule over a 10 μM length. Error bars are the standard deviations calculated from the fluorescence signal of 14 tubules (*N* = 14) from 2 experiments. **D** SMTs after 10 min of incubation with 25 nM scSNX-BAR-GFP, 50 nM scRetromer-mRuby, and 300 μM of RT-D3 or RT-D3 scrambled. **E** Line scan intensity plot of the SMTs indicated by the white box in **D**. **F** SMTs were incubated with 100 nM scSNX-BAR and 300 μM of RT-D3 or RT-D3 scrambled and imaged by confocal microscopy at a frame rate of 0.5 Hz. Red: Texas-Red DHPE. See also Supplementary movie 6.

Yeast *vps5Δ* cells accumulated Vps10-mNeonGreen on fragmented vacuoles labelled with the fluorophore FM4−64 (Fig. 7B, C). A clear recovery of large vacuoles and a localization of Vps10-mNeonGreen on punctate structures was detected when we re-introduced wildtype Vps5 into the knockout cells. In agreement with data published previously we consider these Vps10-mNeonGreen puncta as Golgi elements[19,35,47–49]. Vps5 containing just PX and BAR domains (vps5Δ1-280) rescued neither the sorting defect nor the excessive vacuolar fragmentation of *vps5Δ* cells and Vps10-mNeonGreen accumulated on the vacuolar membranes (Fig. 7B, C). A single amino acid substitution

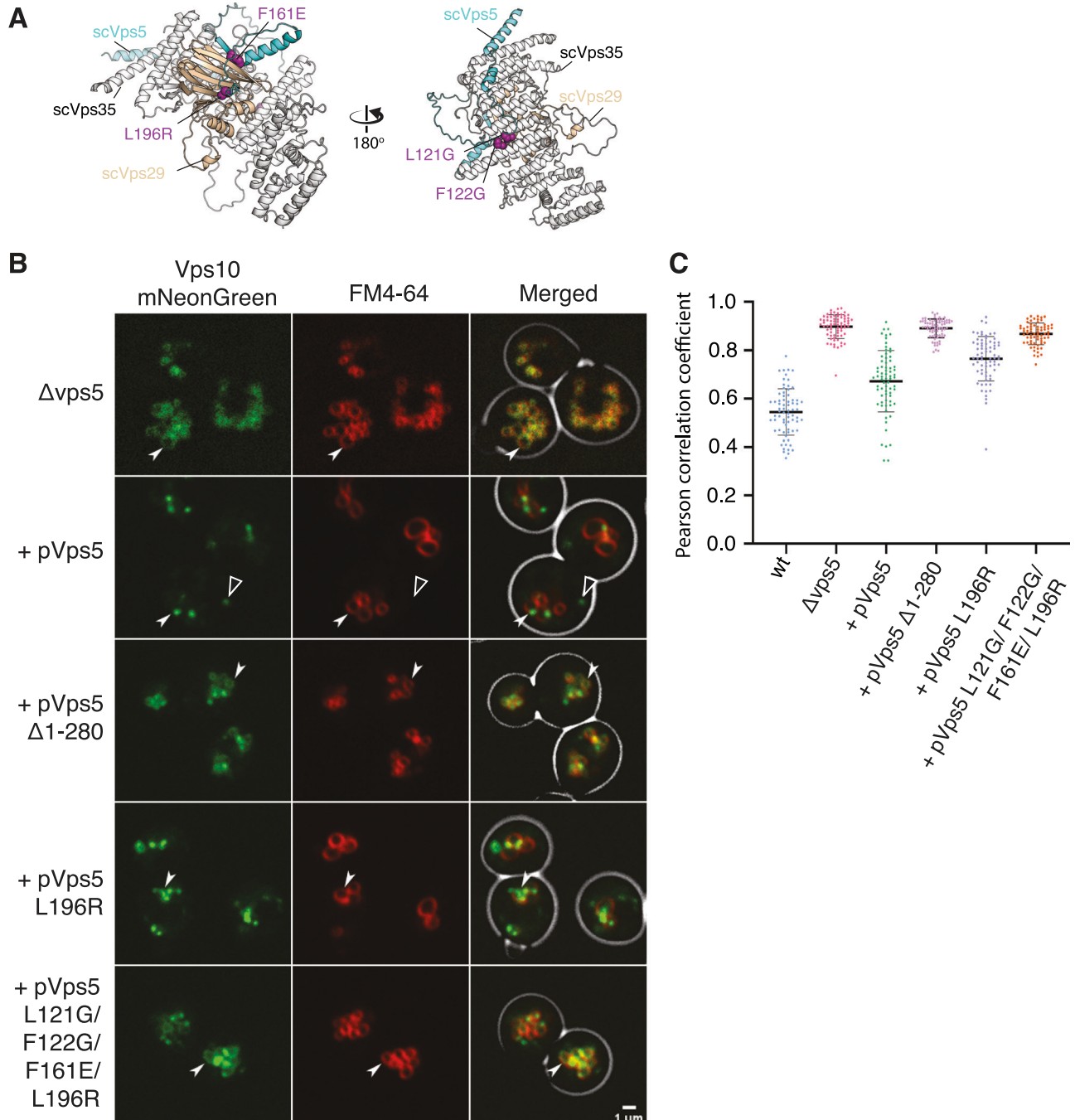

**Fig. 7 | The interaction of Retromer with the Vps5 N-terminal domain is required for Vps10 recycling. A** AlphaFold2 predicted structure of the scVps35–scVps29–scVps5 complex highlighting residues targeted for mutagenesis. **B** Yeast cells (SEY6210) carrying a Vps5 deletion, and expressing a copy of Vam10 (the VAM10 gene is embedded in VPS5 and deleted with it) and Vps10-mNeonGreen were complemented with centromeric plasmids expressing variants of Vps5. These cells were logarithmically grown in SD-URA medium, stained with FM4-64 and analysed by confocal microscopy. Single slices of the images are shown. Scale bar: 1 μm. Arrowheads indicate regions of Vps10 vacuolar localisation, and open arrowheads indicate Vps10 localised to pre-vacuolar comparmtnets. **C** Graph representation showing the Pearson Colocalizastion Coefficient between the vacuolar and Vps10 signals of Vps5 deleted yeast cells shown in **B** quantified using JACoP (Just Another Co-localization Plugin) implemented in ImageJ. Error bars are the standard deviations calculated from ≥70 measurements from 2 experiments.

in the core PL motif, Vps5 (L196R), generated a partial defect in Vps10 recycling and a partial rescue of vacuole fragmentation. The quadruple mutant Vps5 (L121G/F122G/F161E/L196R) known to impair pentameric coat formation, showed a severe Vps10 recycling sorting defect and strong vacuole fragmentation similar to *vps5Δ* cells (Fig. 7B, C). These results corroborate the structural and biochemical data, and demonstrate the relevance of the combined PL, [K/R] Φ and LFa motifs of scVps5 for interacting and functioning with Retromer in vivo.

## Discussion

The Retromer trafficking complex was originally discovered in *S. cerevisiae* as a pentameric complex with the SNX-BAR proteins Vps5 and Vps17. Recent structural studies of both Retromer from *C. thermophilum* and metazoan species have shown that the core Retromer trimer of Vps35–Vps26–Vps29 forms an asymmetric and oligomeric arch-like structure through a series of heterotypic and homotypic interactions, including Vps35 homodimerization at the apex and Vps26

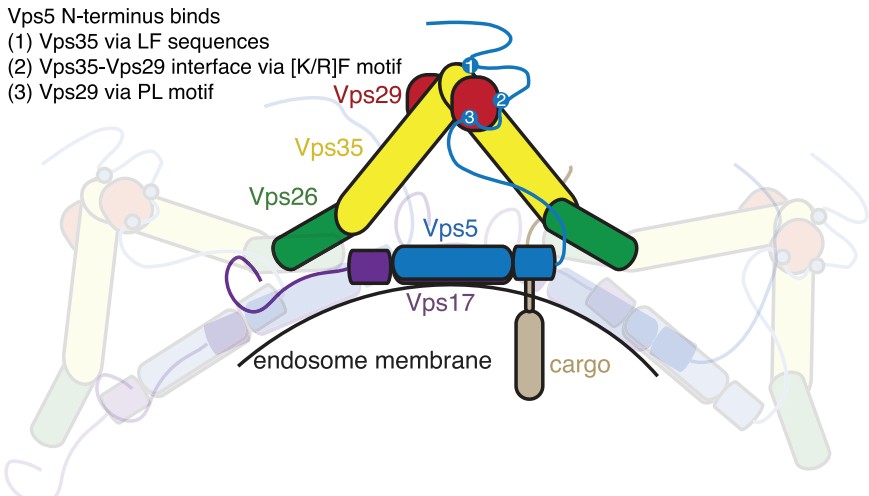

Vps5 N-terminus binds
(1) Vps35 via LF sequences
(2) Vps35-Vps29 interface via [K/R]F motif
(3) Vps29 via PL motif

**Fig. 8 | Proposed model for the assembly of yeast Retromer with Vps5–Vps17 required for vacuolar cargo recycling.** Schematic diagram of the yeast Retromer–Vps5–Vps17 complex and its assembly on the endosomal/vacuolar membrane. The sequences located in the flexible N-terminus of Vps5 include PL and [K/R]ϕ motifs, and additional LF-containing sequences specific to the *Saccharomycotina* subdivision binds to Vps29 and Vps35 sites at the apex of the Retromer arch. This high affinity interaction is required for initial pentameric complex formation prior to subsequent assembly at the membrane surface where Vps26 then forms homodimers on top of the BAR domains of Vps5–Vps17 heterodimers. Blocking the N-terminal Vps5 interaction disrupts Retromer assembly and prevents recycling of cargos such as Vps10 resulting in their accumulation at the vacuole.

homodimers forming direct contacts with the membrane-adjacent SNX protein adaptors[32,33,50]. In the case of the SNX-BAR proteins, Retromer subunit Vps26 assembles atop the surface of the membrane-bound array of PX and BAR domains, with Vps29 located at the distal apex of the Vps35 subunit. However, this model could not account for previous studies in *S. cerevisiae* that have shown the interaction of Retromer with the Vps5–Vps17 proteins requires Vps29 but not Vps26, and also requires the disordered N-terminal domain of Vps5[18,34,35]. This has led to an alternate model for Retromer and SNX-BAR assembly[35].

Our in vitro and in vivo studies of Retromer from both *C. thermophilum* and *S. cerevisiae* reconcile and extend these models of Retromer and SNX-BAR assembly. Our data shows that Vps5 in diverse yeast species contains a conserved Vps29-binding PL motif in the N-terminal unstructured region that is required both for Retromer association in solution and for recruiting Retromer into a functional membrane-assembled complex. In addition, the high affinity binding of yeast Retromer with Vps5 is promoted by two other key interactions. The first involves a [K/R]Φ motif upstream of the PL motif that makes an extended contact with the interface between the Vps29 and Vps35 Retromer subunits. This is conserved across all yeast species we have analysed. A second involves binding of a Leu and Phe containing sequence in Vps5 to the conserved convex surface of the Vps35 α-helical solenoid. This is present in species such as *S. cerevisiae*, but absent in *C. thermophilum*, and the binding mechanism is predicted to be like the recently identified LFa motifs of FAM21 interacting with the human Retromer complex[43]. These findings are completely consistent with both the cryoET structures of membrane assembled Retromer and Vps5[32,33], as well as the cellular studies demonstrating the importance of Vps29 and Vps5 N-terminal regions in complex formation[18,34,35]. Although the membrane-assembled complex involves direct association of Vps26 with the PX and BAR domains as shown by cryoET, the highly extended N-terminal region of Vps5 is capable of high affinity engagement with Vps29 and the C-terminal segment of Vps35 and is essential for their recruitment (Fig. 8). As this work was being prepared, similar findings were reported by Shortill and colleagues, providing further verification of these results[51].

Although previous structural studies by cryoET employed a simplified complex containing a homodimer of Vps5[32,33], we show here that Vps5–Vps17 homodimers can assemble with Retromer into membrane coats with a similar morphology in vitro. In comparison to

Vps5 we found that Vps17 by itself does not stably homodimerize and has little membrane binding or remodelling capacity (Supplementary Fig. 14A–C). Phylogenetic analysis suggests that Vps17 is not present in some early eukaryotes like Archaeplastida and Excavata where Retromer may instead form a complex with a Vps5 homodimer[52]. This raises the question of what role the Vps17 protein plays in Retromer-mediated transport, as it is not required for Retromer interaction and membrane assembly. One possibility is that Vps17 could allow the heteropentameric complex to differentiate between specific cargoes. This idea is at least partly supported by the finding that deletion of Vps17 affects the recycling of Vps10 but not the alternative Retromer cargo Ear1, while Vps5 is required for both[35].

The SNX-BAR family of proteins almost universally possess unstructured N-terminal regions of different lengths and sequences upstream of the structured PX and BAR domains required for membrane interaction and dimeric complex formation[53]. Unlike the PX and BAR domains, the functional roles of these disordered regions are only just starting to be uncovered. The N-terminus of SNX1 and SNX2, the human homologues of Vps5, contain multiple DLF motifs capable of binding to the Retromer cargo adaptor SNX27[54,55]. However, SNX1 and SNX2 appear to lack the PL motif found in yeast Vps5 and are unable to bind human Retromer under similar conditions where Vps5 binds yeast Retromer with high affinity. This likely explains why previous studies have reported either no association[21–23,56,57] (and this study) or only transient association[26,28,29,58–60] between the human SNX-BAR proteins and Retromer complexes. Current data suggests that the metazoan SNX1 and SNX2 proteins most likely associate indirectly with Retromer, potentially through SNX27 to form a cargo handover complex[54,55], although the precise spatial and temporal sequence of events of this are unclear. In addition to SNX1 and SNX2, the yeast SNX-BAR protein Vin1 (also known as YKR078W/Vps501) has been reported to associate with the VARP homolog Vrl1 through its unstructured N-terminus to form an endosomal transport carrier[61]. Although Vrl1 has a PX-like and a BAR domain, the interaction does not rely on the typical BAR domain dimerization. Instead, the unstructured N-terminus of Vin1 is the primary driver of the interaction. An unexpected role of the disordered N-terminus has also been reported in another yeast SNX-BAR protein Mvp1, a protein known to promote Vps1-mediated fission of Retromer – SNX-BAR tubules[62]. In this instance it was shown that the unstructured N-terminus of Mvp1 promotes the formation of

autoinhibited tetrameric assembly[63]. Interestingly, several studies have also shown that some proteins, including BAR domain-containing proteins, can employ combinations of structured and disordered regions together to drive membrane remodelling through molecular crowding of the entropically disordered regions[64–66]. Whether the N-terminal regions of Vps5–Vps17 and their human counterparts play a similar role in promoting membrane bending remains to be tested.

In summary, the findings presented here uncover a previously suspected but molecularly uncharacterised high affinity interaction between Retromer and the N-terminal IDR of Vps5. Notably this involves several motifs that are generally conserved, but also show distinct plasticity in their precise sequences and mechanism of Retromer interaction. This suggests a conserved requirement for this interaction across yeast species, but that the interaction mechanism itself is evolutionarily malleable. This could help explain why in humans the SNX-BAR proteins have apparently lost this Retromer-binding capacity but have instead evolved sequences in their N-terminal domains for binding distinct but functionally related proteins such as SNX27[54,55]. Furthermore, the site in VPS29 that binds to PL-containing motifs has itself been adapted in the human protein to mediate interactions with human specific proteins including the Rab7 GAP TBC1D5[36] and VPS35-related protein VPS35L[5–7]. How such activities have evolved, and how they are regulated across species will be an important question for the future. More generally, Retromer-targeting therapeutics are currently being developed to bind specific Retromer interfaces to stabilise and chaperone its function in endosomal homeostasis[39,67,68], and understanding the precise details of Retromer's scaffolding functions will be essential for assessing the functional impacts of such compounds.

## Methods

### Lipids
The following lipids were purchased from Avanti Polar Lipids (USA): Egg L-alpha-phosphatidylcholine (EPC); 1,2-dioleoyl-sn-glycero-3-phospho-L-serine sodium salt (DOPS); 1,2-dioleoyl-sn-glycero-3-phospho-(1′-myo-inositol-3′-phosphate) (PI3P); 1,2-dioleoyl-sn-glycero-3-phosphoethanolamine-N-(Cyanine 5.5) (Cy5.5 PE); 1-Oleoyl-2-[12-[(7-nitro-2-1,3-benzoxadiazol-4-yl)amino]dodecanoyl]-sn-Glycero-3-Phosphocholine (NBD-PC). 1-palmitoyl-2-oleoyl-sn-glycero-3-phosphocholine (POPC) with 1-palmitoyl-2-oleoyl phosphatidylethanolamine (POPE). Folch fraction I, brain extract from bovine brain was purchased from Sigma-Aldrich (USA). All lipids were dissolved in chloroform. Phosphatidylinositol phosphates were dissolved in chloroform/methanol/water (20:10:1). Texas red DHPE (Thermofisher cat. T1395MP) was purchased as a mixed isomer. The para isomer was separated by thin layer chromatography as previously described[46].

### Molecular biology
For the expression of *C. thermophilum* Retromer in bacteria, full-length cDNA encoding Vps35, Vps26 and Vps29 were cloned into either His-tagged pET28a and GST-tagged pGEX-4T-2 vectors as described previously[32]. For ctVps29, an addition PreScission protease recognition site was engineered in front of the gene for efficient cleavage of GST-tag during purification. In the case of mVps29, the long isoform consists of an additional 5 residues on the N-terminus (starting MAGHR) and was cloned into pGEX-4T-2 vector without any extra linker between the construct and the thrombin recognition site. We found that the GST-tag cleavage was successful in the long isoform Vps29 without the need of extra linker observed in short isoform Vps29 (starting MLVL). cDNA encoding full-length ctVps5 was cloned into His-tagged pET28a and GST-tagged pMCSG10 vectors. Full-length ctVps17 was cloned into pGEX-4T-2 vector, while ctVps17 encoding the PX and BAR domains was cloned into pET28a vector. For the truncation constructs, the DNA sequence encoding ctVps5$_{1-50}$, ctVps5$_{51-70}$, ctVps5$_{61-80}$, ctVps5$_{51-100}$, ctVps5$_{101-150}$, ctVps5$_{1-185}$, ctVps5$_{PX-BAR}$,

ctVps17$_{1-160}$, and ctVps17$_{527-643}$ were cloned into pGEX-4T-2 vector. Site-directed mutagenesis was performed to generate mutant constructs. All constructs were verified using DNA sequencing.

### Cell culture, strains and plasmids
The *Saccharomyces cerevisiae* strains are derived from *SEY6210* (Supplementary Table 3). Vps5 and Pep4 deletions, Vps5 N-terminal point mutations and endogenous tagging of Vps10 was conducted based on the CRISPR-Cas9 system[69]. The vector co-expressing single guide RNA and Cas9 enzyme was co-transformed into yeast cells with double-stranded oligonucleotides as the templates for homologous recombination (Supplementary Table 4). For Vps5 deletion, the template contained 40-bp homology upstream and downstream of its coding sequence. Vps10$^{mNeonGreen}$ was created by a template comprising its C-terminal region with mutated PAM sequence and fused mNeon-Green tag followed by 40-bp homology region after the stop codon. Vam10 was cloned into pRS305 vector using Not1/Sma1 restriction sites to generate an integrative plasmid that was linearized, transformed and integrated ectopically into the auxotrophic selectable marker *LEU2* gene locus[70].

The *VPS5* gene with its native promoter, C-terminally fused HA-tag and ADH1 terminator was cloned into pRS314 centromeric plasmid. A point mutation was introduced in the start codon of *VAM10* (M-to-I) by site-directed mutagenesis, because *VAM10* overlaps with *VPS5*[71]. This exchange does not change the amino acyl residues encoded on the opposite strand for *VPS5* but abolishes *VAM10* expression. For the N-terminal deletion mutant of Vps5 (Δ1-280), residues 838–2025 of *VPS5* were amplified and subcloned into the *pRS314-VPS5-HA* plasmid described above using Nco1/Sma1 restriction sites. The plasmid bearing the PL motif mutation (L196R) was generated using site-directed mutagenesis in the *pRS314-VPS5-HA* plasmid. To abolish all three putative binding sites to Retromer, a synthetic gene fragment of *VPS5* (ordered from GenScript) containing the following mutations—start codon of *VAM10* sequence(M-to-I), L196R, L121G, F122G and F161E—was subcloned into the *pRS314-VPS5-HA* plasmid using Nco1/Stu1 restriction sites.

### Live microscopy of yeast cells
Vacuoles were stained with FM4–64. Cells were grown in SD$^{-URA}$ medium to an OD$_{600}$ between 0.6 and 1.0. The culture was diluted to an OD$_{600}$ of 0.4, and FM4–64 was added to a final concentration of 10 μM from a 10 mM stock in DMSO. Cells were labelled for 60 min, washed three times in fresh media, and then incubated for another 60 min. Cells were concentrated by a brief low-speed centrifugation and placed on a glass microscopy slide overlaid with a thin glass coverslip (#1.5). Z-stacks with a spacing of 0.3 μm were collected on a NIKON Ti2E Yokogawa spinning disk confocal microscope and single slices of images were made.

### Recombinant protein production in bacteria
All the bacterial constructs were expressed in BL21(DE3) competent cells using standard isopropyl 1-thio-β-D-galactopyranoside induction or autoinduction method as described previously[39,72,73]. To obtain the Retromer trimer complex, the GST-tagged Vps29 and Vps35 co-expressed cell pellet were mixed with His-Vps26 and resuspended in lysis buffer containing 50 mM Tris-HCl pH 7.5, 200 mM NaCl, 5% glycerol, 50 μg/ml benzamidine and DNase I before passed through a Constant System TS-Series cell disruptor. Similarly, the GST-tagged ctVps5–ctVps17$_{PX-BAR}$ heterodimer were obtained using co-expression method as described above. For ctVps5 homodimer, the His-tagged construct was used for optimal expression yield. For the GST-tagged full-length ctVps17, approximately 100 g of cell pellet was collected and lysed for each purification due to the poor expression yield. For all constructs, the lysate was clarified by centrifugation and the supernatant was loaded onto either Talon® resin (Clonetech) or glutathione

sepharose (GE healthcare) for initial purification. To obtain the correct stoichiometry ratio of the Retromer complex, both Talon® resin and glutathione sepharose (GE healthcare) were applied. Removal of the GST-tag from mVps29, ctVps29 and ctVps5 was performed using on-column cleavage overnight at 4 °C. The flow-through containing complex was further purified using size-exclusion chromatography (SEC) equilibrated with a buffer containing 50 mM Tris-HCl pH 7.5, 200 mM NaCl, 2 mM β-ME. To obtain ctRetromer–ctVps5 and the pentameric complexes, the purified ctRetromer was mixed with either purified ctVps5 or ctVps5–ctVps17$_{PX-BAR}$ complex in 1 to 1 ratio for at least 1 h at 4 °C. To avoid the non-specific cleavage by the residual proteases such as thrombin, a tablet of protease inhibitor cocktail was added into the complex mixture. To remove the unbound fractions, the complex mixture was further purified using Superose 6 increase 10/300 GL column using the SEC buffer described above.

Expression and purification of the GST-tagged truncation and mutation constructs including hSNX1$_{1-139}$, ctVps5$_{1-50}$, ctVps5$_{51-70}$, ctVps5$_{61-80}$, ctVps5$_{51-100}$, ctVps5$_{51-100}$ PL74AA, ctVps5$_{51-100}$ RKL/AAG, ctVps5$_{101-150}$, ctVps5$_{1-185}$, ctVps5$_{PX-BAR}$, ctVps17$_{1-160}$, scVps5$_{70-128}$ and scVps5$_{70-128}$ mutant (L90R, I91R, L94R, I101R, L104R, L121R and F122R) was performed using the same buffer and method described above. In brief, the clarified lysate was loaded onto glutathione sepharose (GE healthcare) and eluted using elution buffer containing 50 mM Tris-HCl pH 7.5, 200 mM NaCl, 5% glycerol, 20 mM glutathione reduced and 50 μg/ml benzamidine. The flow-through was further purified using SEC as the final purification step. To avoid the degradation, the constructs containing N-terminal disordered parts of Vps5 and Vps17 were used in fresh and experiments were finished within 3 days after purification.

### Recombinant protein production in yeast

TAP-tagged scRetromer and scSNX-BAR complexes were extracted from yeast as previously described[45,74]. Briefly, a 50 mL preculture of cells was grown over night to saturation in YPGal medium. The next day, two 1 L cultures in YPGal were inoculated with 15 mL of preculture and grown for 20 h to late log phase (OD$_{600}$ = 2 to 3). All following steps were performed at 4 °C. Cells were pelleted and washed with 1 pellet volume of cold RP buffer (Retromer purification buffer: 50 mM Tris pH 8.0, 300 mM NaCl, 1 mM MgCl$_2$, 1 mM PMSF, Roche complete protease inhibitor). Pellets were either processed immediately or flash-frozen in liquid nitrogen and stored at −80 °C. For cell lysis, the pellet was resuspended in one volume of RP buffer and passed through a French press (One shot cell disruptor, Constant Systems LTD, Daventry, UK) at 2.2 Kpsi. 1 mg DNase I was added to the lysate followed by a 20 min incubation on a rotating wheel. The lysate was precleared by centrifugation for 30 min at 45,000 × g in a Beckman JLA 25.50 rotor and cleared by a 60 min centrifugation at 150,000 × g in a Beckman Ti60 rotor. The cleared supernatant was passed through a 0.2 μm filter and transferred to a 50 mL Falcon tube. 1 mL IgG bead suspension (GE Healthcare, cat 17-0969-01) was washed three times with RP buffer and added to the supernatant. After 60 min incubation on a rotating wheel, beads were spun down and washed 3 times with RP buffer. 250 μg of purified HIS-TEV protease from *E. coli* was added to the beads. After 30 min incubation at 4 °C, beads were centrifuged, the supernatant containing purified Retromer subcomplex was collected and concentrated on a 100 kDa cutoff column (Pierce™ Protein Concentrator PES, 100 K MWCO). The concentrated protein fraction was re-diluted in RP buffer and reconcentrated 3 times. This final step allowed for removal of TEV protease and a high enrichment for intact complexes. Proteins were concentrated to ~2 mg/mL, aliquoted in 10 μL fractions and flash-frozen in liquid nitrogen. Proteins were stored at −80 °C and used within 3 months. Thawed aliquots were used only once.

### Differential scanning fluorimetry

Thermal unfolding experiments were carried out to confirm the protein constructs applied in this study were properly folded or not. In brief, 0.2 mg/ml of purified ctVps26, ctVps29, ctVps29–ctVps35, ctRetromer, ctVps5 and ctVps5–ctVps17$_{PX-BAR}$ was prepared with the addition of 5× final concentration of SYPRO orange dye (Life Science). The sample was loaded into the 96-well plate and the relative fluorescence units were measured from 25 °C to 85 °C using the ROX dye calibration setting at 1 °C increments using a ViiA7 real-time PCR instrument (Applied Biosystems). Experiments were performed with four replicates, and $T_m$ was calculated using Boltzmann sigmoidal in Prism version 10.2.0 (GraphPad Software).

### Isothermal titration calorimetry

The binding affinities between Retromer and various SNX-BAR constructs were determined using a Microcal PEAQ ITC (Malvern) at 25 °C. All the proteins used for ITC experiments were prepared in buffer containing 50 mM Tris-HCl pH 7.5, 200 mM NaCl, 2 mM β-ME. To map the precise binding region between ctVps5 and ctRetromer, 300 μM of GST-tagged ctVps5 N-terminal truncation constructs including GST-ctVps5$_{51-50}$, GST-ctVps5$_{51-100}$, GST-ctVps5$_{101-150}$ and ctVps5$_{PX-BAR}$ was titrated into 10 μM ctRetromer. The effect of cyclic peptides on ctVps5 and ctRetromer interaction was performed by titrating 200 μM of full-length ctVps5 or GST-ctVps5$_{51-100}$ into 10 μM ctRetromer or 10 μM ctVps29 containing either 100 μM of RT-D3 or RT-D1 L7E mutant cyclic peptides. The importance of K/R]Φ or the PL motifs were carried out by titrating 10 to 15 μM of ctRetromer with various truncation constructs (200 μM–350 μM) and peptides (340 μM–400 μM) of ctVps5 including GST-ctVps5$_{51-100}$ PL74AA mutant, GST-ctVps5$_{51-100}$ RKL/AAG mutant, GST-ctVps5$_{51-70}$, GST-ctVps5$_{61-80}$, ctVps5$_{53-80}$ and ctVps5$_{71-80}$ peptides. To examine if the N-terminal region of ctVps17 also involved in binding with ctRetromer, 300 μM of GST-ctVps17$_{1-160}$ was titrated into 10 μM ctRetromer. In parallel, 300 μM of GST-ctVps5$_{1-185}$ was titrated into 10 μM ctRetromer or hVps29 for the comparison analysis. The effect of cyclic peptides on ctVps29 were carried out by titration 100 μM of RT-D3 or RT-D1 L7E into 10 μM of ctVps29. In all cases, the ITC titration sequence consisted of an initial 0.4 μl (not used in data processing) followed by 12 serial injections of 3.22 μl each with 180 s intervals. The resulting titration data were integrated with Malvern software package by fitting and normalized data to a single-site binding model, yielding the thermodynamic parameters $K_d$, ΔH, ΔG and −TΔS for all binding experiments. The stoichiometry was refined initially, and if the value was close to 1, then N was set to exactly 1.0 for calculation. Data presented are the mean of triplicate titrations for each experiment.

### GST pull-down assays

For the GST pull-down assay, 1 nmol GST-tagged ctVps5, ctVps5–ctVps17$_{PX-BAR}$ heterodimer, ctVps5$_{1-50}$, ctVps5$_{51-100}$, ctVps5$_{1-150}$, and GST alone were mixed separately with 2 nmol of ctRetromer or ctDimer (ctVps29–ctVps35 subcomplex). For the pull-down between GST-ctVps5 and ctVps17$_{PX-BAR}$ heterodimer and ctRetromer, either 8 molar excess of RT-D3 or equivalent volume of DMSO were added to examine the effect of PL motif on pentameric complex assembly. All reaction mixtures were incubated for 2 h at 4 °C before added to fresh glutathione Sepharose pre-equilibrated with buffer containing 50 mM Tris-HCl pH 7.5, 200 mM NaCl, 5% glycerol, 0.5 mM TCEP and 0.1% IGEPAL CA-630 for another 1 h at 4 °C. GST pull-down assay of scVps5 and hRetromer was performed similar to the one described above with some modifications. The purified hRetromer at 2 nmol was mixed with 1 nmol GST-tagged hFam21$_{R19-R21}$, scVps5$_{70-128}$ LF native, scVps5$_{70-128}$ LF mutant or GST alone separately overnight at 4 °C before added to fresh glutathione Sepharose pre-equilibrated with buffer as mentioned above. In all cases, beads were washed five times with the equilibration buffer and bounds proteins were analysed by SDS-PAGE gels.

## Liposome binding and tubulation assays with negative stain electron microscopy

The control PC/PE liposome was prepared by mixing POPC with POPE in a 90:10 molar ratio. In brief, 34.2 μl of POPC at 13.2 mM (10 mg/ml) was mixed with 3.6 μl of POPE at 13.9 mM (10 mg/ml), to a total volume of 500 μl chloroform in a mini round bottom flask. The lipid film was dried down on the walls under N2 stream and left overnight in a vacuum desiccator. Similarly, the Folch I-PtdIns(3)P lipid film was prepared by adding 50 μl of 10 mg/ml Folch fraction I with 47.9 μl of di-C16 PtdIns(3)P, to a total volume of 500 μl chloroform. The lipid film was dried down in the same way as PCPE lipid film. To form multi-lamellar vesicles (MLVs), the lipid film was hydrated with protein SEC buffer containing 50 mM Tris-HCl pH 7.5, 200 mM NaCl, 2 mM β-ME for 30 min and then 10 cycles of rapid freeze-thaw process. To form small unilamellar vesicle (SUVs), the MLVs were extruded 21 times using 400 μm pore size membrane. The liposome pelleting assay was conducted in a total volume of 60 μl comprising equal volume of 10 μM ctRetromer–ctVps5 complex, ctVps26, pentameric complex, ctVps5 or ctVps17 alone and SUVs in the presence of DMSO, RT-D3 or RT-L4. The reaction mixture was incubated at room temperature for 30 min followed by 60,000 rpm (rotor TLA100) for 35 min at 4 °C. The supernatant and pelleted fractions were then carefully separated and analyzed by SDS-PAGE.

Tubulation assays were performed similar to the protocol described previously[32]. In brief, 0.5 mg/ml final concentration of Folch I-PtdIns(3)P liposome was incubated with 4 to 5 μM of protein or protein complex indicated including ctRetromer, ctRetromer–ctVps5, ctRetromer–ctVps5 with 3 molar excess RT-D3, ctRetromer–ctVps5–ctVps17_{PX-BAR} pentameric complex or ctVps17 alone for 30 min at room temperature in protein SEC buffer before spotted on a carbon/formvar-coated copper mesh grid. Sample on the grid was washed once with SEC buffer followed by three times MilliQ water prior negative stained by 2% uranylacetate for 30 s. The grid was imaged and analyzed on a FEI Tecnai F30 G2 or JEOL JEM1011 transmission electron microscope.

## Supported membrane tubes (SMTs)

Supported membrane tubes were generated as described[46]. Briefly, glass coverslips were first washed with 3 M NaOH for 5 min and rinsed with water before a 60 min treatment with piranha solution (95% $H_2SO_4$/30% $H_2O_2$ 3:2 v/v). Coverslips were rinsed with water and dried on a heat block at 90 °C. Coverslips were then silanized with 3-glycidyloxypropyltrimethoxysilane (Catalogue no. 440167, Sigma) for 5 h under vacuum, rinsed with acetone and dried. Polyethylene glycol coating was done by placing the coverslips in a beaker containing PEG400 (Sigma) at 90 °C for 60 h. Coverslips were washed with distilled water and stored for up to 2 months at room temperature in a closed container.

To generate supported membrane tubes, lipids were mixed from 10 mg/mL stocks in a glass vial and diluted to a final concentration of 1 mg/mL in chloroform. The same lipid mix was used throughout this study (5% PI(3)P, 15% DOPS, 0.1% fluorescent lipid tracer (Cy5.5 PE or Texas red DHPE), 79.5% egg-PC). Lipids were then spotted (typically 1 μL, corresponding to about 1 nmol) on the coverslips and dried for 30 min under vacuum. The coverslip was mounted on an IBIDI 6-channel μ-slide (μ-Slide VI 0.4. IBIDI, catalog no: 80606). Lipids were hydrated for 15 min with buffer (PBS) and SMTs were generated by injecting PBS into the chamber using an Aladdin Single-Syringe Pump (World Precision Instruments, model n°. AL-1000) at a flow rate of 1,5 mL/min for 5 min. SMTs were left to stabilize without flow for 5 min before the start of the experiment. Protein stocks (typically 1–2 μM) were first diluted in PBS and then injected in the chamber at a flow rate of 80 μL per minute. Cyclic peptides were added to the proteins from a 30 mM stock in DMSO to a final concentration of 300 μM. the protein/ peptide mix was incubated for 1 min at room temp (23 °C) before being added to the SMTubes. Tubes were imaged with a NIKON Ti2 spinning disc confocal microscope equipped with a 100 × 1.49 NA objective.

## Crystallization and data collection

All crystals were grown by the hanging-drop vapor diffusion method at 20 °C. For co-crystallization of ctVps5_{71−80} peptide, sixfold molar excess of ctVps5_{71−80} peptide was added to the purified mVps29 protein to a final protein concentration of 11 mg/ml. Best quality crystals were obtained by mixing 1 μL of protein with 1 μL of well solution containing 40 mM KH_2PO_4, 16% PEG8000 and 5% glycerol at room temperature. Crystals grew after 13 days were cryoprotected in reservoir solution with additional 16% glycerol. X-ray diffraction data were carried out at the Australian Synchrotron MX2 beamline at 100 K.

## Crystal structure determination

Data collected from the Australian Synchrotron were indexed and integrated by AutoXDS and scaled using Aimless (version 0.7.4)[75]. Both structures were solved by molecular replacement using Phaser (version 2.5.6)[76] with native mVps29 structure as the initial model template. Both structures contain two copies of mVps29 in asymmetric unit with the electron density of peptide clearly visible. The refinement was performed using Phenix (version 1.19.2) with inspection of resulting model in Coot (version 0.8.9.2) guided by $F_o$–$F_c$ difference maps[77]. Molprobity was used to evaluate the quality geometry of the refined structure[78]. Data collection and refinement statistics are summarized in Supplementary Table 1. Molecular figures were generated using PyMOL (version 2.5.5) and structural comparison was performed using DALI and Foldseek[79,80]. Sequence conservation was mapped on the structure with ConSurf server[81].

## Cryo-electron tomography grid preparation and data acquisition

The ctRetromer–ctVps5–ctVps17_{PX-BAR} pentameric complex for CryoET data acquisition was incubated at a final concentration of 6.5 μM purified complex with 0.6 mg/ml of freshly prepared Folch I-PtdIns(3)P liposomes for 30 min at room temperature in SEC buffer. Prior to blotting, 10 nm Protein A-gold fiducials were added to the protein-liposome tubulation mixture in 1:5 ratio. The blotting was carried out by loading 3.2 μL of the reaction mixture on a glow-discharged holey carbon grid (CF-2/1–3 C, protochips) and plunge-frozen in liquid nitrogen cooled liquid ethane using a Vitrobot (Mark IV, Thermo Fisher Scientific).

Imaging acquisition was performed on a 200 kV Talos Arctica (Thermo Fisher Scientific) transmission electron microscope fitted with a Falcon 3EC direct electron detector (Thermo Fisher Scientific) at the University of New South Wales, Australia. A total of 73 dose-symmetric tilt series were collected with tilt range ± 60°, 3° angular increment and defoci between −3.5 μm and −5.0 μm. The acquisition magnification was 45,000× with a pixel size of 3.24 Å. Total dose of each tilt series was ranged from 120 to 130 e/A². Data collection parameters are summarized in Supplementary Table 2.

## Tomogram reconstruction

Assembly of tomograms from raw movies was done with IMOD 4.12.45 package[82]. Gain-reference corrected movies from the microscope were aligned, summed and filtered according to the deposited electron dose using aligframes (IMOD 4.12.45) to generate sorted tilt series used for tomogram reconstruction. Per-tilt defocus estimation was done on non-dose weighted stacks in ctfplotter[83]. The stacks further aligned using fiducials-based, 2D-CTF corrected and used for weighted back projection tomogram reconstruction by etomo package (IMOD 4.12.45). For visualisation only, 8-times binned tomograms were processed with deep learning denoising and wedge restoration package IsoNet[84].

## Subtomogram averaging

Subtomogram alignment was done as previously[33] using the subTOM package (https://github.com/DustinMorado/subTOM/releases/tag/v1.1.4), and custom and dynamo-package[85] MATLAB scripts. Initial subtomogram positions were calculated by manual tracing of coated tubules in bin4 tomograms using UCSF Chimera (1.16) and PickParticle plugin[86] and placing subtomogram positions in the middle of the coat with orientation normal to the tube surface at sampling frequency of 2 pixels (~ 30 Å). Twenty-nine coated tubules, including both well and poorly ordered (see example in Fig. 5D), were selected from 6 tomograms. The initial average was generated from the seeded positions. A ~ 120° (by circumference) sector of the coat layer in the initial average, masked by lengthwise cut tube shape, was twofold symmetrized and subjected to an iterative alignment and averaging routine until convergence, with the gradual refinement of angular and translational search. The resulting cryoEM map contained arch-like structures that were smeared in the narrow direction of arches. Inspection of tomogram positions in space (Chimera PlaceInSpace plugin[86] indicated that this is due to failure of subtomograms to reliably lock on arches in z-direction of tomograms on poorly ordered but not well-ordered tubules. Further STA was done only with ordered tubules after removing subtomogram duplicates converged to the same positions by distance threshold and using per tubule cross-correlation threshold to remove subprograms that lost membrane alignment. The cleaned dataset contained ~600 particles in 3 tubules split in odd-even datasets and refined independently in bin2 tomograms. The highest low-pass filter used in the STA was 38 Å, and FSC between the two half maps inside the alignment mask reached 32 Å at 0.143 (Supplementary Fig. 12E). For the final map, the half maps were summed, sharpened with B-factor of −2500 and filtered to the measured resolution (Supplementary Fig. 12F).

## Model building

The PDB models of the ctRetromer arch (dimer of dimers of ctVps39/ctVps35, (7BLR) and dimer of ctVps26 (7BLQ)) were placed manually in densities and then fitted as ridged bodies using the sequential fit command in Chimera at resolution 32 Å. The initial placement of the arch PDB was decided by comparing the fit cc-values for its two orientations (0° and 180° rotated). The gap in cross-correlation scores was apparent for all four arch densities in the map (0.7 vs 0.9).

## AlphaFold predictions

Structural predictions of Retromer and SNX-BAR proteins were performed using Alphafold2-multimer[40,41] implemented within the freely accessible ColabFold pipeline[42]. For each modeling experiment, ColabFold was executed using default settings where multiple sequence alignments were generated with MMseqs2. To map the exact binding regions between ctRetromer, ctVps5 and ctVps17, we first conducted the modelling using full-length proteins. The high-confidence binding region in Vps5 was then mapped and cross-checked with our biochemical data prior of generating a more "focused" model containing shorter sequences. The focused models including ctVps35$_{407-869}$−ctVps29−ctVps5$_{1-180}$ and ctVps35−ctVps26−ctVps17$_{527-643}$ complexes were performed in combination with AMBER energy minimization to optimize amino acid stereochemistry. In each run, we validated the quality of models using the PAE, the prediction confidence measures including pLDDT and interfacial ipTM scores as well as the alignment of the multiple structures generated using pymol. Sequence conservation was mapped on the predicted structure with ConSurf server[81].

## Mass photometry

Molecular mass measurement was performed using a Refeyn OneMP mass photometer (Refeyn Ltd) following manufacturer's instructions.

Prior to the experiment, calibration was performed using protein standards with known molecular weight (i.e., 66, 132 and 480 kDa) in buffer containing 50 mM Tris-HCl pH 7.5, 200 mM NaCl, 2 mM β-ME. Next, 10 μl of indicated protein at a final protein concentration of 100 nM–50 nM was loaded onto the mass photometer, and 1000 frames were recorded. For the molecular mass measurement at the low ionic strength condition, all proteins were dialyzed into low salt buffer containing 50 mM Tris-HCl pH 7.5, 40 mM NaCl, 2 mM β-ME prior of recording. The precise molecular mass was calculated using the calibration standards as reference.

## Reporting summary

Further information on research design is available in the Nature Portfolio Reporting Summary linked to this article.

## Data availability

Crystal structure of hVps29−ctVps5$_{71-80}$ peptide complex has been deposited at the Protein Data Bank with accession code 8FUD. Source Data file is available with this paper. All other raw data related to this study is available from the corresponding authors on request. Source data are provided with this paper.

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

## Acknowledgements

We acknowledge the use of the Australian Microscopy and Microanalysis Research Facility at the Center for Microscopy and Microanalysis at The University of Queensland. We also thank the use of the University of Queensland Remote Operation Crystallization and X-ray (UQROCX) facility for the crystallization experiments. X-ray data was collected at the Australian Synchrotron MX2 beamline with the support from the beamline scientists. CryoET data was collected at the Electron Microscope Unit, Mark Wainwright Analytical Centre at University of New South Wales. This work is supported by funds to BMC from the Australian Research Council (ARC) (DP160101743), National Health and Medical Research Council (NHMRC) (APP1156493), and Bright Focus Foundation (A2018627S). BMC was supported by an NHMRC Senior Research Fellowship and NHMRC Investigator Grant (APP1136021; APP2016410). AM was supported by grants from the Swiss National Science Foundation (SNSF) (179306 and 204713) and European Research Council (ERC) (788442), and SRC by EMBO ALTF 240-2023.

## Author contributions

Initial Concept: B.M.C., N.L. and K.C. Concept development: all authors. Biochemistry, ITC, molecular modelling, structure determination and analysis: K.C., N.L., Q.G. and B.M.C. S.M.T. assays and yeast experiments: N.G. and S.R.C. CryoET studies. K.C., V.A.T., O.K., J.R. and N.A. Data analysis and figure generation: K.C., V.A.T., N.G., S.R.C., O.K. and B.M.C. Funding acquisition: A.M. and B.M.C. Supervision: K.C., N.A., A.M. and B.M.C. Writing – 1st draft: K.C., N.G., A.M. and B.M.C. Writing – review & editing: all authors.

## Competing interests

The authors declare no competing interests.
