## [Transparent Peer Review file · Nature Communications]

Molecular basis for the assembly of the Vps5-Vps17 SNX-BAR proteins with Retromer

Corresponding Author: Professor Brett Collins

Version 0:

Reviewer comments:

Reviewer #2

(Remarks to the Author)

The evolutionarily conserved Retromer complex plays a pivotal role in controlling cargo recycling and sorting at the endosome, and its conservation spans from humans to the simplest single-celled eukaryotes. Previous cryoET studies of Retromer–Vps5 have elucidated a pseudo-helical coat on membrane tubules, where dimers of the Vps26 subunit bind to Vps5 membrane-proximal domains. In their study, Chen et al. demonstrated that VPS5 binds to Retromer via three regions within its unstructured N-terminal domain, including [K/R]Φ, PL, and LF motifs. While the biochemical data presented in the study are robust, I have several major concerns.

1. The ITC assay depicted in Figure 4, PL, [K/R]Φ, and LF mutations on VPS5 are necessary.
2. The SMT assay shown in Figure 6, please incorporate peptides containing [K/R]Φ and LF motifs
3. Move Figure 5 to supplemental figures as it contribute minimally to the paper. Alternatively, increase the resolution to obtain more information.
4. The manuscript has numerous writing mistakes. The authors need to do a better job to avoid these mistakes.

Reviewer #3

(Remarks to the Author)

The study by Chen et al. describes the molecular basis for the interaction of the SNX complex with the retromer complex in yeast. The authors show that retromer binds via Vps29 to the unstructured N-terminal domain of Vps5, which is required to recruit retromer to membrane tubules. This is surprising as Vps29 is far away from the membrane in the assembled complex. The authors further confirm the conservation of the general architecture of the Vps5-Vps17 interaction with Retromer using cryo-tomography, which has been previously analyzed with the Vps5 dimer – a complex that is not functional in vivo. Apparently, the Vps17 interaction with cargo explains its function in vivo as the authors speculate.

The authors provide a very detailed interaction analysis using biophysical interaction tools and AI-tools to predict interfaces, and provide further evidence using crystallography. Using this, they identify a region in Vps5 that is clearly involved in retromer recruitment. They combine this with a peptide as a competitor (RT-D3) that binds Vps29, thus manipulating the recruitment on lipid tubes. Overall, the study combines an impressive set of data. I have some minor issues that should be considered during the revision:

1. The authors introduce their macrocyclic peptides like RT-D3 on page 5, but this comes for me out of nowhere. I recommend a reasonable introduction and also an image, what the peptide does. Figure S2 should be thus integrated into the main figures and the rationale of using this tool should be explained sufficiently.
2. Figure 7 is hard to digest initially as the authors do not show single slices but projections of the entire cell, which makes the images of vacuoles and dots very crowded. I have a few recommendations: The outline of the cells should be indicated in the images. I know that this is their style, but phenotypes are difficult to follow. This figure needs arrows to show the Vps10 localization, and the mutants need to be quantified as it is not clear whether Vps10 is proximal to the vacuole or inside in most mutants. From my understanding, yeast deletions are written *vps5(delta)*.

Reviewer #4

(Remarks to the Author)

This work studies the molecular interactions between the retromer complex and sorting nexins proteins in *Chaetomium thermophilum*. In particular, the authors test the specific role of the VPS29 and VPS35 proteins in inducing interactions with the SNX Bar proteins to form a pentameric complex. The manuscript follows up on a previous report from the same lab where cryo-tomography was used to describe the organization of retromers-VPS5 complexes bound to membranes.

The present manuscript, in many aspects, appears as additional independent pieces of information as compared with the previous report generated from the same team. It thus does not necessarily gather a full single strong story, which lower its impact, unfortunately.

In general, most likely for making the manuscript as concise as possible, some pieces of information are partially missing, especially for the readers that are not in the present research field. For instance, Figures 2D and 2E are commented and presented with little minimal information. Further description would be valuable. The RT-D3 peptide could also be described to highlight its previous characterization. Hence the manuscript is difficult to follow, at places. The proteins considered vary depending on the assay presented and the rationale for these choices is rarely explained. Also the protein nature vary from ct to cerevisiae and mammals as well with no clear explanation.

The most questionable assay would be the pull-downs from Figure 1. In Figures 1.C and 1.D, some of the observations or bands appearing on the gel are not commented.

Why GST-ctVPS5 appears as two bands? Are multimers expected? Also there is no indication of expected molecular mass for comparison. Do they correspond to GST-ctVPS5 and truncated VPS5?

A lot of degradation bands are visible and not commented in the manuscript.

The ct retromer complexes as well as the ctVPS35-29 complex do not appear stoichiometric on the gel. Is it expected? What is the expected stoichiometry of the complexes? Why is the band corresponding to ctVPS35 much stronger than the other ones?

Eventually, the authors should perform mass spectrometry to analyze a bit better the nature of the proteins in specific bands. Also have the authors tested that individual proteins from the retromer complex, when purified individually remain properly folded? Have the authors tested that VPS26 itself does not interact with vps5, using a pull down? What about the VPS26-VPS35 and VPS26-VP29 combination? Would they interact with VpS5? Besides, have any similar experiments been performed using VPS17 only or in complex with VPS5?

For crystallization, it would be informative to know which constructs have been screened.

Also, given Alpha fold analysis highlights interactions between Vps5 and Vpd29 and VPs5 and VPS35, did any crystallization tests have been carried out using VPS35 and appropriate domains of VPS5? If not, why have the authors prioritized the VPS29-VPS5 contact? Also, is it relevant to draw conclusions on a complex formed using a mammalian protein (from mouse) with a counterpart from fungi, in terms of conservation of properties? Can the authors comment on this? Indeed, it is specified that human SNX1 and 2 behave differently from ct proteins.

Concerning the cryo-tomography and sub-tomogram averaging, the result remains very similar to the previous published structure obtained from homodimers of VPS5. Can the authors specify how they made the tubule diameter more regular? Was it induced experimentally or did they only pick the ones with similar diameters? Also can they display a statistical representation of the tubule diameter distribution? What would be the proportion of tubules over round shaped vesicles in their samples?

To get a final higher resolution, why a higher magnification was not chosen (ie a lower pixel size than the 3.2 Å used here)? Also why only 600 particles were used for STA? Was it difficult to reproduce the tubulation? Was it only obtained once? Do the authors plan on improving the resolution for a follow up publication?

Instead of picking a sub volume enclosing a full portion of the tubes, have the authors attempted to pick smaller boxes only displaying portion of a bilayer decorated with protein?

Concerning the dynamics of tubule coat formation assay, it is not clear why the authors perform those experiments using *Cerevisiae* proteins instead of ct proteins? Can they perform the same experiments with ct proteins, to make the manuscript more homogeneous?

minor remarks:

In Figure 2D and E, what are the dimensions of liposomes? Because Bar domains are curvature sensitive, that would be informative. The control of liposomes could be performed by cryo-EM or DLS.

In the introduction the authors stress the differences between retromer-SNX Bar protein association between higher and lower eukaryotes, they should also insist on why it is interesting to focus their study on ct retromers and why they chose to study proteins from this organism, in particular.

Reviewer #5

(Remarks to the Author)

The Retromer is a conserved trimeric complex of Vps26, Vps29 and Vps35 proteins that functions in cargo retrieval from endosomes. Studies in yeast indicate that the Retromer functions in concert with a heterodimer of two PX-BAR domain-containing membrane bending/tubulating proteins Vps5-Vps17. While interactions between the Retromer and PX-BAR proteins have been extensively studied in yeast and other mammalian homologues, specific interactions between individual components and their impact on the Retromer coat formation on membranes remain unclear and is an important area of investigation.

The same group had previously reported a structure of the Retromer with a Vps5 homodimer, which revealed that Vps26 can bind Vps5. This was surprising because earlier studies indicate that Vps29 is required for the association of the retromer with the Vps5-Vps17 heterodimer. Here, the authors reconcile this discrepancy by structural analysis of the Retromer with the Vps5-Vps17 heterodimer. They identify an unstructured N-terminal region containing a PL motif that binds Vps29. Furthermore, the authors establish a novel [K/R] Φ motif that makes extensive contacts with the Vps29-Vps35 interface. They further show that the above interactions are required for Retromer functions in cargo retrieval in yeast.

This study is a significant addition to the existing knowledge. The following comments only aim to seek clarifications and suggest experiments that would enhance the appeal and impact of this already exciting work.

Specific comments:

Liposome sedimentation and SMrT assays indicate that the presence of RT-D3 impacts recruitment of the Retromer-SNX-BAR complex to the membrane. This result implies that Retromer binding to SNX-BAR facilitates membrane binding of the Retromer-SNX-BAR complex, which is not intuitive given that the Retromer does not directly bind membranes and that the RT-D3 peptide should just interfere with Retromer-SNX-BAR interaction. Perhaps, the authors could discuss this result and test binding of SNX-BAR alone and in presence of the Retromer to further evaluate the basis of this effect. These experiment could also serve as controls for evaluating the effect of the RT-D3 and scrambled peptides.

Data showing recruitment of Retromer-SNX-BAR complex on SMrTs (Fig. 6C) reveal differences in the recruitment rates between Retromer and the SNX-BAR. Given that both complexes are introduced simultaneously, can the authors comment on these differences? Additionally, it would be helpful to present the data from RT-D3 and scrambled in a single plot to allow direct comparison of the membrane recruitment rates.

SMrT assays show that the Retromer-SNX-BAR complexes constrict the underlying tubule. Considering SNX-BAR proteins are known to tubulate membranes, what might be the basis for such constriction? The authors could report the dimensions of tubules drawn from liposomes and indicate whether these dimensions are thinner than the starting dimensions of tubes on SMrT, thus explaining the constriction.

Minor Comments:

The data in Fig. 6A does not align with Movie S4. The authors might be referring to Movie S5 rather than S4. Please check.

The timestamp in Movie S6 is cropped. The authors could rectify this movie.

Version 1:

Reviewer comments:

Reviewer #1

(Remarks to the Author)

1. The ITC assay depicted in Figure 4, PL, [K/R] Φ , and LF mutations on VPS5 are necessary.

Response to the authors: One of the key findings of this study is the identification of three motifs—[K/R] Φ , PL, and LF—essential for VPS5 binding to the Retromer complex. Truncations cannot replace site-specific mutagenesis on these critical residues. The suggested ITC experiments are necessary to make the conclusion.

2. The SMT assay shown in Figure 6, please incorporate peptides containing [K/R] Φ and LF motifs.

We appreciate the suggestion. However, we believe the use of the macrocyclic peptide RT-D3 (and the scrambled RT-D3 negative control) are sufficient in this instance to demonstrate the importance of the PL motif in recruiting scRetromer to the membrane.

Response to the authors: Same as 1. Experiments with site-specific mutations are absolutely needed to support the authors' main conclusion.

Reviewer #2

(Remarks to the Author)

The authors addressed all my questions and added additional experimental details to the figures. I therefore have no further requests and recommend the study for publication.

Reviewer #3

(Remarks to the Author)

The authors have performed additional significant experiments and added comments to clarify and answer the different concerns on their manuscript.

The presented work has thus gained relevance and is now fitted to be published in nature communications.

This will bring additional insights and knowledge to the field.

I thereby thank the authors for improving their manuscript.

Reviewer #4

(Remarks to the Author)

I appreciate the authors' efforts in addressing my queries. The point that the rates of association appear different because the proteins are tagged with different fluorophores can easily be addressed by fitting the data to an exponential rise function and normalizing the data to the fitted plateau value. This would be a nice addition for the reader to evaluate if the rates of association are similar or different. Also, showing the control data set (in the absence of the R3D3 peptide) in the same plot would be useful for comparison. But these are just suggestions and I am otherwise satisfied with the responses provided.

Version 2:

Reviewer comments:

Reviewer #1

(Remarks to the Author)

The authors have addressed all my concerns

REVIEWER COMMENTS

Reviewer #1 (Remarks to the Author):

The evolutionarily conserved Retromer complex plays a pivotal role in controlling cargo recycling and sorting at the endosome, and its conservation spans from humans to the simplest single-celled eukaryotes. Previous cryoET studies of Retromer–Vps5 have elucidated a pseudo-helical coat on membrane tubules, where dimers of the Vps26 subunit bind to Vps5 membrane-proximal domains. In their study, Chen et al. demonstrated that VPS5 binds to Retromer via three regions within its unstructured N-terminal domain, including [K/R]Φ, PL, and LF motifs. While the biochemical data presented in the study are robust, I have several major concerns.

1. The ITC assay depicted in Figure 4, PL, [K/R]Φ, and LF mutations on VPS5 are necessary.

We thank the reviewer for this suggestion. We believe that the ITC experiments shown in **Figures 4E and 4F** of the original manuscript clearly demonstrate the importance of both the PL and [K/R]Φ motifs in ctRetromer binding. First, the truncation constructs of ctVps5, specifically ctVps5₅₁₋₇₀ (lacking the PL motif) and ctVps5₆₁₋₈₀ (lacking the [K/R]Φ motif), both show very weak binding to ctRetromer. Data in **Figure 4F** indicate that substituting the PL motif in ctVps5 to Alanine reduces its binding affinity to ctRetromer by 8-fold.

However, to further support these findings, we performed ITC experiments of ctRetromer against ctVps5 peptides of different lengths. While the binding of ctVps5₅₃₋₈₀ peptide (which contains both PL and [K/R]Φ motifs) is comparable to full-length ctVps5, the peptide ctVps5₇₁₋₈₀ lacking residues 53 to 68 where the [K/R]Φ motif is located exhibits very weak binding to ctRetromer. We have now included the new ITC data showing the binding of ctVps5₅₃₋₈₀ and ctVps5₇₁₋₈₀ with ctRetromer in the revised manuscript as **Figure S10A**, along with the following description in the result section.

“Similarly, the short ctVps5₇₁₋₈₀ peptide contains mainly the PL motif also reveals very weak binding to ctRetromer compares to the long ctVps5₅₃₋₈₀ peptide (**Fig. S10A**)”

For the convenience of the Editor and reviewer, we have also attached below the new **Figure S10A**.

Figure R1 (new Fig. S10A). ITC measurements show that ctRetromer binds strongly to the ctVps5₅₃₋₈₀ peptide (long), which contains both [K/R] Φ and PL motif, compared to the PL motif only ctVps5₇₁₋₈₀ peptide (short). ITC graph shows the integrated and normalized data fit with a 1 to 1 binding ratio.

Regarding the LF motif found in scVps5 and spVps5, we believe the data shown in **Figures 6 and 7** of the original manuscript have demonstrated the avidity effect of both motifs in scRetromer binding. For clarity, we have also created a new **Figure S10D** to illustrate that the LF motif found in scVps5 is predicted to bind to a region of scVps35 that is remarkably like the hFam21 repeat 20 binding regions. As shown in our recently published paper¹, these LF motifs on their own bind weakly to Retromer. For the convenience of the Editor and reviewer to follow, we have also attached new **Figure S10D** below.

Figure R2 (new Fig. S10D). Superimposition of the scVps35-scVps29-scVps5 model with the hVps35-hVps29-hFam21R19-R20 model highlights the conservation of the Vps35 binding site. For clarity, only the C-terminal region of Vps35 and the key binding residues are shown.

2. The SMT assay shown in Figure 6, please incorporate peptides containing [K/R] Φ and LF motifs.

We appreciate the suggestion. However, we believe the use of the macrocyclic peptide RT-D3 (and the scrambled RT-D3 negative control) are sufficient in this instance to demonstrate the importance of the PL motif in recruiting scRetromer to the membrane.

3. Move Figure 5 to supplemental figures as it contributes minimally to the paper. Alternatively, increase the resolution to obtain more information.

In our opinion the CryoET data should stay in the main figure. This data is important as it clearly demonstrates that the physiologically relevant heterodimeric Vps5-Vps17 complex can recruit Retromer and generate tubules. We acknowledge that the resolution of the cryoET data is relatively low, but it still allows us to confidently conclude that there is a distinct difference in morphology to the previously reported Vps5-Retromer tubules². Collecting higher resolution would provide only limited additional insight but would require a significant investment of time and resources that we feel is outside the scope of this study.

4. The manuscript has numerous writing mistakes. The authors need to do a better job to avoid these mistakes.

We have carefully revised the manuscript and corrected any errors we could find. However, without any specific examples we are not sure how else to do a “better job”.

Reviewer #2 (Remarks to the Author):

The study by Chen et al. describes the molecular basis for the interaction of the SNX complex with the retromer complex in yeast. The authors show that retromer binds via Vps29 to the unstructured N-terminal domain of Vps5, which is required to recruit retromer to membrane tubules. This is surprising as Vps29 is far away from the membrane in the assembled complex. The authors further confirm the conservation of the general architecture of the Vps5-Vps17 interaction with Retromer using cryo-tomography, which has been previously analyzed with the Vps5 dimer – a complex that is not functional in vivo. Apparently, the Vps17 interaction with cargo explains its function in vivo as the authors speculate.

The authors provide a very detailed interaction analysis using biophysical interaction tools and AI-tools to predict interfaces, and provide further evidence using crystallography. Using this, they identify a region in Vps5 that is clearly involved in retromer recruitment. They combine this with a peptide as a competitor (RT-D3) that binds Vps29, thus manipulating the recruitment on lipid tubes. Overall, the study combines an impressive set of data. I have some minor issues that should be considered during the revision:

1. The authors introduce their macrocyclic peptides like RT-D3 on page 5, but this comes for me out of nowhere. I recommend a reasonable introduction and also an image, what the peptide does. Figure S2 should be thus integrated into the main figures and the rationale of using this tool should be explained sufficiently.

We thank the reviewer for this valuable comment. Retromer binding cyclic peptides applied in this study including RT-D3, RT-D3 scramble, RT-D1 (L7E) mutant and RT-L4 were identified from *in vitro* RaPID (random nonstandard peptide integrated discovery) screening and were described in our previous paper³. For clarification, we have added a sentence in the first part of the result section to introduce the cyclic peptides. Additionally, we have generated a new **Figure 2B** to schematically illustrate the effect of the cyclic peptides applied in this study. We have also generated additional ITC binding data in the revised **Figure S4B** to show that RT-D3 binds strongly to ctVps29, but the RT-D1 (L7E) mutant does not. For the convenience of the Editor and reviewer to follow, we have also attached new **Figure 2B** and **Figure S4** below.

Figure R3 (new Fig. 2B). Schematic diagram showing the cyclic peptides applied in the ITC and liposome pelleting assay. RT-D3 binds strongly to Vps29 and inhibits interactions of other PL motif-containing proteins. RT-D1 (L7E) is a mutant peptide that does not bind Vps29 and acts as a negative control. RT-L4 acts as a molecular staple that binds to the Vps26–Vps35 interface and does not affect Vps29 from binding to PL motif-containing proteins.

Figure R4 (new Fig. S4A and S4B). Binding of previously identified macrocyclic peptides to ctVps29. (A) Schematic diagram of the four different previously identified³ macrocyclic peptides that applied in this study. RT-D3 contains PL motif (highlight in red) capable of binding to Vps29 and inhibits PL motif-containing proteins such as Vps5 from binding. RT-D1 (L7E) and RT-D3 scrambled are used as control peptides that are unable to interact with Retromer/Vps29 due to loss of the core PL sequence. RT-L4 binds to the Vps26–Vps35 interface. (B) ITC measurements showing the binding between ctVps29 and cyclic peptides RT-D3 or RT-D1 (L7E) mutant. ITC graph shows the integrated and normalized data fit with a 1 to 1 binding ratio.

2. Figure 7 is hard to digest initially as the authors do not show single slices but projections of the entire cell, which makes the images of vacuoles and dots very crowded. I have a few recommendations:

We thank the reviewer for the comments. We have modified **Figure 7B** and included an additional panel **Fig. 7C** in the revised manuscript according to the following questions.

(i) The outline of the cells should be indicated in the images. I know that this is their style, but phenotypes are difficult to follow.

The cell outlines are shown in the merged panels.

(ii) This figure needs arrows to show the Vps10 localization.

We have included arrowheads to indicate the regions of overlap between Vps10-mNeonGreen (green) and the vacuole (red), or when it is localised to pre-vacuolar Golgi compartments in the Vps5-rescued control.

(iii) The mutants need to be quantified as it is not clear whether Vps10 is proximal to the vacuole or inside in most mutants.

We have now included quantitation of Vps10-mNeonGreen localisation to the vacuole using ImageJ in new **Fig. 7C**. The images (single cells were selected using ROI tool) were averagely projected (top to bottom of the vacuolar red signal). Then JACoP plugin ([doi:10.1111/j.1365-2818.2006.01706.x](https://doi.org/10.1111/j.1365-2818.2006.01706.x)) was used to quantify the Pearson Colocalization Coefficient between the vacuolar signal and Vps10 signal. The values are plotted as scatter dot plot and analysed by t test using GraphPad Prism. The results reflect the observations.

As a summary for the reviewers, the main observations were:

- $\Delta vps5$ shows a homogeneous localization of Vps10 on the vacuole membrane.
- Plasmid with wildtype Vps5 have punctate Vps10 pre-vacuolar and Golgi localization.
- Plasmid with Vps5 N-terminal deletion shows localization of Vps10 on the vacuole membrane.
- Plasmid with Vps5 PL mutation shows an intermediate phenotype of Vps10, with partial localization on the vacuolar membrane.
- Plasmid with Vps5 triple mutation shows a strong effect, similar to that of N-terminal deletion.

From my understanding, yeast deletions are written *vps5*(italics)delta.

Thank you, we have corrected this in the manuscript.

For the convenience of the Editor and reviewer to follow, we have also attached revised **Figure 7B** and **7C** as below.

Figure R5. The interaction of Retromer with the Vps5 N-terminal domain is required for Vps10 recycling. (B) Yeast cells (SEY6210) carrying a Vps5 deletion, and expressing a copy of Vam10 (because the VAM10 gene is embedded in VPS5 and deleted with it) and Vps10-mNeonGreen were complemented with centromeric plasmids expressing variants of Vps5. These cells were logarithmically grown in SD^{-URA} medium, stained with FM4-64 and analysed by confocal microscopy. Single slices of the images are shown. Scale bar: 1 μ m. Arrowheads indicate regions of Vps10 vacuolar localisation, and open arrowheads indicate Vps10 localised to pre-vacuolar compartments. (C) Graph representation showing the Pearson Colocalization Coefficient between the vacuolar and Vps10 signals of Vps5 deleted yeast cells shown in (B) quantified using JACoP (Just Another Co-localization Plugin) implemented in ImageJ.

Reviewer #3 (Remarks to the Author):

This work studies the molecular interactions between the retromer complex and sorting nexin proteins in *Chaetomium thermophilum*. In particular, the authors test the specific role of the VPS29 and VPS35 proteins in inducing interactions with the SNX Bar proteins to form a pentameric complex. The manuscript follows up on a previous report from the same lab where cryo-tomography was used to describe the organization of retromers-VPS5 complexes bound to membranes.

The present manuscript, in many aspects, appears as additional independent pieces of information as compared with the previous report generated from the same team. It thus does not necessarily gather a full single strong story, which lower its impact, unfortunately.

We believe our study answers a significant question in the field of endosomal trafficking regarding the mechanism of interaction (or lack thereof) between Retromer and SNX-BAR proteins across evolution. Our study shows that N-terminal sequences in yeast Vps5 orthologues (that are not found in mammalian SNX-BAR homologues) mediate high affinity Retromer binding for pentameric complex assembly. More specifically it unambiguously resolves an important potential disagreement in the field between previous structural findings and cell-based data⁴. The work clearly explains why Vps29 deletion in yeast leads to dissociation of Retromer from the Vps5-Vps17 complex⁴, even though when assembled on membranes the Vps29 subunit lies distally to the Vps5-Vp17 membrane layer.

1. In general, most likely for making the manuscript as concise as possible, some pieces of information are partially missing, especially for the readers that are not in the present research field. For instance, Figures 2D and 2E are commented and presented with little minimal information. Further description would be valuable.

We conducted the liposome pelleting assay in **Figures 2D and 2E** (now 2E and 2F) to confirm whether the PL motif of ctVps5 plays a critical role in recruiting ctRetromer to the membrane or not. The liposome pelleting assay in our manuscript was designed based on previous published data, which demonstrated that Retromer alone only binds to the membrane in the presence of cargo-adaptors such as Vps5² and SNX3⁵. Using this knowledge, we performed the liposome pelleting assay of ctRetromer–ctVps5 complex in the presence and absence of the Vps29 blocking macrocyclic peptide RT-D3. We found as expected that ctVps5 was unable to recruit ctRetromer to the membrane in the presence of the inhibitory peptide. For clarity, we have added new sentences in the results section to provide better description of the background and the data obtained. This text is included below for clarity.

“Next, we examined the membrane recruitment of ctRetromer by ctVps5 using liposome-binding experiments, measuring the fraction of proteins that are either co-pelleted together or remain in the supernatant. To test the importance of the PL-motif interaction we used the competitive cyclic peptide RT-D3 which blocks the binding site on Vps29, or the control RT-L4 which binds to the interface of Vps35–Vps26 and is not expected to affect ctVps5 interaction³⁹ (Figs. 2B, 2D and 2E). In control experiments, ctVps5 itself bound to membranes composed of Folch I lipids supplemented with the phosphoinositide PtdIns3P which binds the ctVps5 PX domain, and this was not affected by either peptide. The ctRetromer complex was robustly recruited to PtdIns3P-containing liposomes in the presence of ctVps5 in agreement with previous results (Figs. 2D, 2E and S6A)³². The cyclic peptide RT-L4 did not affect liposome recruitment of ctRetromer by ctVps5 as expected, whereas RT-D3 significantly reduced the amount of ctRetromer in the pelleted fraction with ctVps5 (Figs 2D and 2E). We still observed the formation of membrane tubules by ctVps5 in the presence of RT-D3, suggesting that blocking the PL motif interaction does not prevent the BAR-domain protein from forming tubules (Fig S6B). In agreement with the solution binding studies, these results show that interaction of the ctVps5 PL motif with Vps29 is important for ctVps5-mediated membrane recruitment of the ctRetromer complex.”

2. The RT-D3 peptide could also be described to highlight its previous characterization. Hence the manuscript is difficult to follow, at places.

We thank the reviewer for raising this question. We have addressed the same question above as requested by Reviewer 1. In summary, the cyclic peptides applied in this study were identified from the *in vitro* RaPID (random nonstandard peptide integrated discovery) screen published previously³. We have generated a new **Figure 2B** in the revised manuscript to schematically illustrate the effects of the cyclic peptides on Retromer used in this study. In addition, we have included a new **Figure S4B** to show that RT-D3 binds strongly to ctVps29 but not the case of RT-D1 (L7E) mutant.

3. The proteins considered vary depending on the assay presented and the rationale for these choices is rarely explained. Also the protein nature vary from ct to cerevisiae and mammals as well with no clear explanation.

We apologise if this has led to any confusion and have tried to provide additional modifiers in the text to explain this more clearly.

To clarify here for the reviewers, in this work we are primarily focused on the unique signatures of yeast Vps5 that leads to high affinity Retromer interactions in contrast to mammalian SNX-BAR proteins and Retromer. Proteins from the two different yeast species (the thermophilic yeast *Chaetomium thermophilum*, and the commonly studied yeast *Sacchomyces cerevisiae*) were used in different experiments for both practical and scientific reasons. We primarily used *C. thermophilum* proteins for biochemical and structural studies based on their relative ease of bacterial expression and purification, and because these proteins were previously established in the Collins lab for cryoET analyses of the membrane assembled complexes². One exception is the crystallographic study of the ctVps5 PL motif-containing peptide, where we screened both ctVps29 and mouse Vps29 and obtained co-crystals with the mouse homologue for structure determination (**Fig. 3E-J**).

For the cellular experiments, we examined the functional impacts of Vps5 interaction with Retromer in *S. cerevisiae* as the strains and genetic approaches are well established in the Mayer lab. Similarly, the small-scale purification and SMTube assays and cell-based studies were performed using *S. cerevisiae* proteins because these methods and constructs are also well established in the Mayer lab⁶. Indeed, an important reason for establishing the collaboration between the Mayer and Collins labs was because of the distinctly complementary systems we each had in place.

While some of these assays might appear straightforward, optimising them for a new set of proteins from another organism is far from trivial. For instance, optimizing the SMTube assay requires substantial effort to manage unspecific adsorption to the surfaces, re-titrate the conditions for all the subunits used to make sure that we are in a range where coat growth is dependent on Retromer–SNX-BAR interaction etc. Similarly, for the cell-based experiments, where years of experience with *S. cerevisiae* proteins have provided functional fluorescent protein fusions for all the used components and suitable conditions for using them for microscopic analyses of sorting. In contrast, none of this is currently possible in *C. thermophilum*, but these proteins provide a more convenient system for the structural studies. But most importantly, as we are looking at interactions that we propose are conserved, testing different aspects of the Retromer–SNX-BAR proteins in related organisms is not only justifiable

but also provides direct confirmation of the conserved nature of this phenomenon in different yeast species.

4. The most questionable assay would be the pull-downs from Figure 1. In Figures 1.C and 1.D, some of the observations or bands appearing on the gel are not commented. Why GST-ctVps5 appears as two bands? Are multimers expected? Also there is no indication of expected molecular mass for comparison. Do they correspond to GST-ctVPS5 and truncated VPS5? A lot of degradation bands are visible and not commented in the manuscript.

In our experience, we always observe slight nonspecific degradation of ctVps5 constructs during its expression and purification. The GST-ctVps5 construct of interest includes a large portion of intrinsically disordered sequence that leads to some proteolytic susceptibility, but this does not significantly impact its ability to bind ctRetromer or assemble with ctVps17 for example. To further address the reviewer's concern, we first confirmed by peptide mass spectrometry that the lower band (at ~47 kDa) observed in GST-ctVps5 samples as shown in **Figure 1C** corresponds to a truncated fragment of GST-ctVps5. This degradation band is always present when purifying the recombinant GST-tagged ctVps5, but it does not compromise the pull-down assays or ITC experiments in any way. We have added a short sentence below and in the **Figure 1** legend of the revised manuscript to clarify the interpretation of this result.

“GST pull-down demonstrating the direct interactions between full-length GST-tagged ctVps5 and either the full ctRetromer complex or the ctVps29-ctVps35 subcomplex. Note, the lower band at ~47 kDa corresponds to a truncated GST-ctVps5 fragment that is present after affinity purification. (D) GST pull-down showing GST-ctVps5 unstructured N-terminal IDR (residues 1-150) is required and sufficient to bind the ctVps29-ctVps35 subcomplex. Black arrowheads indicate bands of full-length GST-ctVps5 constructs. For clarity, brown arrowheads indicate the ctVps35 bands captured in the pull-downs.”

For the convenience of the reviewers, we have also attached revised **Figures 1C** and **1D** as below.

Figure R6. *Vps5 interacts directly with Retromer via its unstructured N-terminus. (C) GST pull-down demonstrating the direct interactions between full-length GST-tagged ctVps5 and either the full ctRetromer complex or the ctVps29-ctVps35 subcomplex. Note, the lower band at ~47 kDa corresponds to a truncated GST-ctVps5 fragment that is present after affinity purification. (D) GST pull-down showing GST-ctVps5 unstructured N-terminal IDR (residues 1-150) is required and sufficient to bind the ctVps29-ctVps35 subcomplex. Black arrowheads indicate bands of full-length GST-ctVps5 constructs. For clarity, brown arrowheads indicate the ctVps35 bands captured in the pull-downs.*

5. The ct retromer complexes as well as the ctVPS35-29 complex do not appear stoichiometric on the gel. Is it expected? What is the expected stoichiometry of the complexes? Why is the band corresponding to ctVPS35 much stronger than the other ones? Eventually, the authors should perform mass spectrometry to analyze a bit better the nature of the proteins in specific bands.

Based on our long experience, whether using mammalian, zebrafish or ctRetromer, Vps35 consistently appears slightly more intense than Vps26 and Vps29 on Coomassie stained SDS-PAGE gels (for example as observed in **Fig. 1C**). However, other assays, such as gel filtration and mass photometry confirm that all three subunits are present in 1:1:1 stoichiometric amounts (for example as shown in **Fig. 2C**). For clarity, we have added a gel-filtration chromatogram and the corresponding SDS-PAGE gel of ctRetromer and ctVps35 – ctVps29 subcomplex as the New **Figure S1A** and **S1B** in the revised manuscript and below.

Figure R7. Figure S1. Integrity and purify of ctRetromer and ctVps29-ctVps35 used in this study. (A) Typical size-exclusion chromatogram of ctRetromer and ctVps29 – ctVps35 subcomplex purified using S200 10/300 column. (B) Associated SDS-PAGE gel showing the purity and integrity of ctRetromer and the subcomplex. (C) Differential scanning fluorimetry showing the folding status of the purified ctVps5, ctVps5 – ctVps17_{PX-BAR}, ctRetromer and the associated subunits.

6. Also have the authors tested that individual proteins from the retromer complex, when purified individually remain properly folded? Have the authors tested that VPS26 itself does not interact with vps5, using a pull down? What about the VPS26-VPS35 and VPS26-VP29 combination? Would they interact with VpS5? Besides, have any similar experiments been performed using VPS17 only or in complex with VPS5?

The first question is mostly answered by the fact that each of the individual Retromer subunits (both from *C. thermophilum* and mammalian species) can be purified to homogeneity in soluble form and behave well in various assays including pull-downs, ITC and even crystallisation (as shown in this study and previous papers from the Collins lab and others). However, to further address this query, we performed differential scanning fluorimetry (DSF) assays on various purified samples. As shown in the new **Figure S1C** of the revised manuscript and below, all the proteins involved in the study including ctVps29, ctVps26, ctVps29–ctVps35, ctRetromer, cVps5 and ctVps5–ctVps17_{PX-BAR} showed a typical temperature denaturation curve suggesting they have a globular fold. This new data is also presented below for the reviewers.

Figure R8. Differential scanning fluorimetry showing the folding status of the purified ctVps5, ctVps5–ctVps17_{PX-BAR}, ctRetromer and the associated subunits.

To address the second question regarding whether ctVps26 itself could bind to ctVps5 or not, we performed the liposome pelleting assay under the same conditions as shown in **Figure 2D** of the manuscript. We found that ctVps26 alone is unable to bind and pellet with ctVps5 on the membrane. Consistent with other data in the manuscript, yeast Vps26 only binds weakly to Vps5 in solution, and requires the interaction of Vps35 and Vps29 and the Vps5 N-terminal domain. Once assembled at the membrane, it can subsequently form a dimeric association that can dock onto the Vps5 array as observed in our CryoET data. For the convenience of the editor and reviewers to follow, we have also included the new data in **Figure S6A** in the revised manuscript and below.

F.P. = Folch I + PtdIns(3)P liposome extruded using 0.4 μ m pore-size membrane.

Figure R9. Liposome-pelleting assay of ctVps26 and ctVps5. In the absence of ctVps29 and ctVps35 subunits, ctVps26 alone fails to interact with ctVps5 in the presence of Folch I lipids supplemented with PtdIns(3)P. “S” and “P” indicates unbound supernatant and bound pellet respectively.

Regarding the question of whether ctRetromer binds directly to Vps17, we believe our data presented in **Figs 5A, 5B, 6** and **7** clearly demonstrated that the Vps5 N-terminus is the most important region for binding to ctRetromer. We showed that mutating or blocking this interaction impairs ctRetromer’s ability to form a complex with ctVps5–ctVps17 and be recruited to the membrane. Based on these findings, we believe it is safe to conclude that Vps17 does not play a significant role in Retromer interaction in solution, and likely only provides a support role in organisation of the pentameric complex on the membrane.

7. For crystallization, it would be informative to know which constructs have been screened. Also, given Alpha fold analysis highlights interactions between Vps5 and Vps29 and Vps5 and VPS35, did any crystallization tests have been carried out using VPS35 and appropriate domains of VPS5? If not, why have the authors prioritized the VPS29-VPS5 contact? Also, is it relevant to draw conclusions on a complex formed using a mammalian protein (from mouse) with a counterpart from fungi, in terms of conservation of properties? Can the authors comment on this? Indeed, it is specified that human SNX1 and 2 behave differently from ct proteins.

We screened several constructs of Vps29 (including ctVps29 and mouse Vps29), Retromer and Vps5 fragments for crystallisation, but ultimately, we were only able to report structures of the complexes that produce diffraction quality crystals. For unknown reasons we have found that similar PL-motif containing peptides reported in our previous publication crystallise more readily in complex with mouse Vps29 than with ctVps29³. We believe it is valid to use mouse Vps29 in this study as the PL motif binding surface is highly conserved across yeast and mammalian proteins. We have generated a new **Figure S8** in the revised manuscript and presented below, which highlights the similarity in PL motif binding across the two species. We believe this new supplementary figure can also explain the differences in binding affinity for ctVps29 and mouse Vps29 shown in **Figures 3I** and **3J**. We have also attempted crystallisation experiments with ctVps5 fragments and the ctRetromer complex without success. In the future we could test different ctVps35 constructs (e.g. as done in our recent publication of human Vps35 fragments bound to peptides from the Fam21 protein¹), but this would require significant time and resources and we believe would have little impact on the current paper.

Figure R10. Comparison of the *ctVps5*₇₁₋₈₀ peptide and RT-D3 bound Vps29 models. (A) Cartoon representation of the *ctVps29*–RT-D3 crystal structure (PDB ID: 6XS8), (B) *mVps29*–*ctVps5*₇₁₋₈₀ peptide crystal structure (this study), and (C) *ctVps29*–*ctVps5* AlphaFold2 model. For clarity, only conserved hydrophobic surface of Vps29 and the key interacting residues are shown.

8. Concerning the cryo-tomography and sub-tomogram averaging, the result remains very similar to the previous published structure obtained from homodimers of VPS5. Can the authors specify how they made the tubule diameter more regular? Was it induced experimentally or did they only pick the ones with similar diameters? Also can they display a statistical representation of the tubule diameter distribution? What would be the proportion of tubules over round shaped vesicles in their samples?

We have now gone back and measured the diameters of multiple *ctVps5*–Vps17–Retromer coated tubules across several CryoET micrographs. These measurements are included in the revised manuscript as new **Figures S12C** and **S12D**. This is also shown below as **Figure R11** for the reviewer’s convenience. The tubules exhibit a similar level of variation in their diameter as those previously observed for *ctVps5*–Retromer, with an average outer diameter of 40 nm (compared to ~26.3 nm for the membrane layer). However, as described in the manuscript, there appears to be more heterogeneity in the organisation of the coat itself.

Figure R11. Analysis of Retromer tubules by negative stain EM and cryoEM. (C) Example CryoET micrograph of *ctRetromer*–*ctVps5*–*ctVps17PX*–*BAR* coated tubules at 0° tilt. (D) Histogram showing the diameter of vesicles (left) and tubules (right) generated by the pentameric complex. For the tubules, two different measurements were carried out. The histogram highlighted in orange indicates the measured distance from one side of the

membrane to another. The measurement in light blue indicates the distance from the apex region of the observed protein coat from one side to the other side of the membrane layer. For all measurements, a total of 50 tubules and vesicles were randomly selected from the collected tilt series.

9. To get a final higher resolution, why a higher magnification was not chosen (ie a lower pixel size than the 3.2 Å used here)? Also why only 600 particles were used for STA? Was it difficult to reproduce the tubulation? Was it only obtained once? Do the authors plan on improving the resolution for a follow up publication?

The magnification was chosen to maximise the acquisition efficiency (larger field of view for large pixel size) but does not compromise the attainable resolution (Nyquist limit is 6.3Å). Another practical consideration for us was the limitation of what can be achieved with parallel illumination in the 2-condenser lens system of the Arctica - we are limited in spot size and magnification.

Regarding the number of particles this was deemed sufficient to obtain maps describing the general organisation of the coat. Due to the relatively low numbers of well organised tubules suitable for sub-tomogram averaging, it became time-limiting to pick larger numbers of particles for sub-tomogram averaging. We do not currently plan to collect more data to increase the resolution (for example using a higher voltage microscope with higher magnification) as this would require significant investment in resources and time and provide little additional information. Even if with greater efforts it was possible to improve the data to sub-nanometre resolution it would still not be sufficient to clearly distinguish between Vps5 or Vps17, or to observe the binding of Vps5 N-terminal sequences with Retromer by cryoET.

We selected all particles in the available well-ordered tubules. This modest particle number produced an averaged map with sufficient resolution to discern the arrangement of the coat. The next interpretability milestone would come at ~10Å resolution where secondary structure elements become visible. That however will require a dataset of ~10³ more particles, a large and costly experimental undertaking outside the scope of this paper, likely without providing the necessary discrimination between Vps5 or Vps17 nor or to observe the binding of Vps5 N-terminal sequences with Retromer by cryoET..

10. Instead of picking a sub volume enclosing a full portion of the tubes, have the authors attempted to pick smaller boxes only displaying portion of a bilayer decorated with protein?

A section of the tube was masked for the alignment, not the whole tube slice. We selected alignment masks to be sufficiently featureless and “soft” to prevent masking artifacts and over alignment. This can be verified in **Figure S12F** showing the orthoslices of the final map. Cryo-electron density in the map propagates smoothly outside the masked area with no detectable masking boundaries that may give an impression that entire sections of the tube were masked. We thank the reviewer for highlighting this potentially lack of clarity and have extended the description of the masking approach in the methods

“A ~120° (by circumference) sector of the coat layer in the initial average, masked by lengthwise cut tube shape, was 2-fold symmetrized and subjected to an iterative alignment and averaging routine until convergence, with the gradual refinement of angular and translational search”

11. Concerning the dynamics of tubule coat formation assay, it is not clear why the authors perform those experiments using *Cerevisiae* proteins instead of ct proteins? Can they perform the same experiments with ct proteins, to make the manuscript more homogeneous?

As we mentioned above, many experiments were initially developed for specific a set of proteins from certain organisms. Applying proteins from other organisms requires extensive optimization and may be difficult to interpret. We believe this only becomes an important issue if the outcomes reveal discrepancies between the experiments. In our case, the importance of the PL motif mediated binding is conserved across different yeast species. A similar example has also been published by Klink and colleagues where the biochemistry and CryoET and cryoEM structure of Mon1-Ccz1 complex was solved using proteins from *C. thermophilum* and the cell-based data was performed using proteins from *S. cerevisiae*⁷.

minor remarks:

12. In Figure 2D and E, what are the dimensions of liposomes? Because Bar domains are curvature sensitive, that would be informative. The control of liposomes could be performed by cryo-EM or DLS.

All the liposomes were prepared by extrusion using 400 nm pore-size membrane. For the convenience of the Editor and reviewer to follow, we have also included a negative staining image of Folch I liposome supplemented with PtdIns(3)P in both the revised manuscript, **Figure S6B**, and below.

Figure R12. Representative negatively stained image of the 400 nm extruded unilamellar vesicles used in this study.

13. In the introduction the authors stress the differences between retromer-SNX BAR protein association between higher and lower eukaryotes, they should also insist on why it is interesting to focus their study on ct retromers and why they chose to study proteins from this organism, in particular.

As mentioned above the use of *C. thermophilum* proteins (and judicious comparisons with *S. cerevisiae* proteins) in this study was initially based on our previous CryoET structure of

membrane assembled ctRetromer–ctVps5². An initial impetus to our work was the study by Suzuki et al., that suggested there might be potential discrepancies between the structure and interactions observed in yeast cells⁴. Our current study provides a conclusive explanation for these apparent discrepancies and shows that in fact the structural models and cellular data on Retromer and Vps5 interaction are entirely consistent with each other.

Reviewer #4 (Remarks to the Author):

The Retromer is a conserved trimeric complex of Vps26, Vps29 and Vps35 proteins that functions in cargo retrieval from endosomes. Studies in yeast indicate that the Retromer functions in concert with a heterodimer of two PX-BAR domain-containing membrane bending/tubulating proteins Vps5-Vps17. While interactions between the Retromer and PX-BAR proteins have been extensively studied in yeast and other mammalian homologues, specific interactions between individual components and their impact on the Retromer coat formation on membranes remain unclear and is an important area of investigation.

The same group had previously reported a structure of the Retromer with a Vps5 homodimer, which revealed that Vps26 can bind Vps5. This was surprising because earlier studies indicate that Vps29 is required for the association of the retromer with the Vps5-Vps17 heterodimer. Here, the authors reconcile this discrepancy by structural analysis of the Retromer with the Vps5-Vps17 heterodimer. They identify an unstructured N-terminal region containing a PL motif that binds Vps29. Furthermore, the authors establish a novel [K/R]Φ motif that makes extensive contacts with the Vps29-Vps35 interface. They further show that the above interactions are required for Retromer functions in cargo retrieval in yeast.

This study is a significant addition to the existing knowledge. The following comments only aim to seek clarifications and suggest experiments that would enhance the appeal and impact of this already exciting work.

We thank reviewer for the comments. We have addressed the specific questions below.

Specific comments:

1. Liposome sedimentation and SMT assays indicate that the presence of RT-D3 impacts recruitment of the Retromer-SNX-BAR complex to the membrane. This result implies that Retromer binding to SNX-BAR facilitates membrane binding of the Retromer-SNX-BAR complex, which is not intuitive given that the Retromer does not directly bind membranes and that the RT-D3 peptide should just interfere with Retromer-SNX-BAR interaction. Perhaps, the authors could discuss this result and test binding of SNX-BAR alone and in presence of the Retromer to further evaluate the basis of this effect. These experiments could also serve as controls for evaluating the effect of the RT-D3 and scrambled peptides.

In our model, yeast Retromer is indeed only able to associate with membranes through binding with the yeast Vps5–Vps17 complex² (or Snx3⁵). (Similarly, mammalian Retromer is only able to associate with membranes indirectly via other SNX adaptors or Rab-GTPases). Consistent with this, yeast Retromer is not efficiently recruited to the yeast Vps5–Vps17 coated liposomes or SMTs when the cyclic peptide RT-D3 is present to inhibit the Vps5 interaction. This peptide

does *not* affect membrane association of Vps5–Vps17 heterodimers or Vps5 homodimers, however. For example, as shown in **Figure 2D**, and new **Figure S6B** of the revised manuscript, ctVps5 can bind and tubulate liposomes in the presence of RT-D3 as expected. While outside the scope of the current work, it is likely that these tubules are coated by a loosely organized array of dimeric BAR-domain containing proteins.

Figure R13. Negatively stained images of ctRetromer–ctVps5 complex in the presence (right image) and absence (middle image) of cyclic peptide RT-D3. The image on the left shows a typical Folch I liposome supplemented with PtdIns(3)P, extruded using a 400 nm pore-size membrane.

In addition, the SMTube assay in **Figures 6A** and **6B** of the manuscript clearly demonstrate that yeast SNX-BARs bind the tubules in the presence of RT-D3. Only the recruitment of Retromer to the SNX-BAR coated tubules is suppressed by the cyclic peptide. Furthermore, as shown in **Figure 6F**, we have confirmed that the RT-D3 peptide on its own does not interfere with SMTube binding of the Vps5–Vps17 complex in the absence of Retromer.

2. Data showing recruitment of Retromer-SNX-BAR complex on SMrTs (Fig. 6C) reveal differences in the recruitment rates between Retromer and the SNX-BAR. Given that both complexes are introduced simultaneously, can the authors comment on these differences? Additionally, it would be helpful to present the data from RT-D3 and scrambled in a single plot to allow direct comparison of the membrane recruitment rates.

We thank the reviewer for the question. Retromer and SNX-BARs carry different fluorescent proteins that have different quantum yields. As a result, even with similar rates, the fluorescence signals have different slopes on a y-axis in which the primary signals had been plotted. An argument that the rate of SNX-BAR and Retromer recruitment is similar can be drawn from **Figure. 6B**, where the kymographs show that the Retromer- and SNX-BAR zones are extended essentially in parallel.

We have replotted the graphs grouping the fluorophores shown below and in **Figure S13** of the revised manuscript. Note that the apparent lower amount of SNX-BAR-GFP recruitment in the presence of RT-D3 is due to the lack of organised coat formation with Retromer.

Figure R14. Quantification of the scSNX-BAR-GFP and scRetromer-mRuby recruitment to SMTs. The graph shows the replot of Figure 6C with scSNX-BAR-GFP (left) and scRetromer-mRuby (right) grouped according to the presence of cyclic peptide RT-D3 or RT-D3 scrambled. Note that the lower amount of scSNX-BAR-GFP recruitment in the presence of RT-D3 was due to the lack of organised coat formation with Retromer.

3. SMrT assays show that the Retromer-SNX-BAR complexes constrict the underlying tubule. Considering SNX-BAR proteins are known to tubulate membranes, what might be the basis for such constriction? The authors could report the dimensions of tubules drawn from liposomes and indicate whether these dimensions are thinner than the starting dimensions of tubes on SMrT, thus explaining the constriction.

Constriction is driven by the formation of the SNX-BAR coat and Retromer oligomerization that impose a narrower diameter to the tubule. We have previously quantified the starting radius of the tubes in the SMT assay⁶. They range from 40 nm to 100 nm in radius, which is indeed larger than the diameter measured for the tubules formed by Retromer on liposomes (22 to 34 nm; now presented in Fig. R11D). The diameter of the Retromer coat measured in the SMT assay using dynamin and lipid fluorescence as a reference is 19.1 +/- 0.6 nm⁶ and thus in a similar range as in tubules formed from liposomes and analysed by EM.

Minor Comments:

4. The data in Fig. 6A does not align with Movie S4. The authors might be referring to Movie S5 rather than S4. Please check.

Apologies. **Movies S4** and **S5** were labelled the wrong way around and have now been fixed in the revised paper. Text has been revised on Page 10 to reflect these changes.

5. The timestamp in Movie S6 is cropped. The authors could rectify this movie.

We believe the timestamp is not cropped in **Movie S6**. This may appear to be the case if being viewed with Quicktime player while the cursor is active in the viewing window. This can lead to the control bar of the window obscuring the timestamp.

References:

1. Guo, Q. et al. Structural basis for coupling of the WASH subunit FAM21 with the endosomal SNX27-Retromer complex. *Proc Natl Acad Sci U S A* **121**, e2405041121 (2024).
2. Kovtun, O. et al. Structure of the membrane-assembled retromer coat determined by cryo-electron tomography. *Nature* **561**, 561-564 (2018).
3. Chen, K.E. et al. De novo macrocyclic peptides for inhibiting, stabilizing, and probing the function of the retromer endosomal trafficking complex. *Sci Adv* **7**, eabg4007 (2021).
4. Suzuki, S.W., Chuang, Y.S., Li, M., Seaman, M.N.J. & Emr, S.D. A bipartite sorting signal ensures specificity of retromer complex in membrane protein recycling. *J Cell Biol* **218**, 2876-2886 (2019).
5. Leneva, N., Kovtun, O., Morado, D.R., Briggs, J.A.G. & Owen, D.J. Architecture and mechanism of metazoan retromer:SNX3 tubular coat assembly. *Sci Adv* **7**(2021).
6. Gopaldass, N. et al. Retromer oligomerization drives SNX-BAR coat assembly and membrane constriction. *EMBO J* **42**, e112287 (2023).
7. Klink, B.U. et al. Structure of the Mon1-Ccz1 complex reveals molecular basis of membrane binding for Rab7 activation. *Proc Natl Acad Sci U S A* **119**(2022).

REVIEWER COMMENTS

Reviewer #1 (Remarks to the Author):

1. The ITC assay depicted in Figure 4, PL, [K/R] Φ , and LF mutations on VPS5 are necessary.

Response to the authors: One of the key findings of this study is the identification of three motifs—[K/R] Φ , PL, and LF—essential for VPS5 binding to the Retromer complex. Truncations cannot replace site-specific mutagenesis on these critical residues. The suggested ITC experiments are necessary to make the conclusion.

We again thank the reviewer for the concern. To address the question regarding the site-specific mutagenesis of the [K/R] Φ motif, we have generated the mutant of ctVps5₅₁₋₁₀₀ containing R50A, R53A, R58A, R59A, K63A, K65A and L67G substitution, namely, RKL/AAG mutant. Using ITC, we found GST-ctVps5₅₁₋₁₀₀ RKL/AAG mutant failed to bind to ctRetromer compared to the WT construct under the same conditions. This result correlates with other binding data in the manuscript, demonstrating that both [K/R] Φ and PL motifs are essential for the binding to Retromer. We have now included the new ITC data showing the binding of ctVps5₅₁₋₁₀₀ RKL/AAG mutant with ctRetromer in the revised manuscript as Figure 4G, along with the following descript in the result section.

“the ctVps5₅₁₋₁₀₀ construct with the [K/R] Φ motif (R50A, R53A, R58A, R59A, K63A, K65A and L67G) mutated to alanine and glycine resulted a complete loss of binding (**Fig. 4G**). The AlphaFold model and biochemical studies thus show a role of both the [K/R] Φ and PL motifs in associating with Retromer.”

For the convenience of the Editor and reviewer, we have also attached below the new **Figure 4G**.

Figure R1 (new Fig. 4G). ITC measurements show that ctRetromer failed to bind to GST-ctVps5₅₁₋₁₀₀ RKL/AAG mutant compares to the wild-type. ITC graph shows the integrated and normalized data fit with a 1 to 1 binding ratio.

Regarding the LF motif observed in scVps5, we first generated the scVps5₇₀₋₁₂₈ construct containing the LF motif and the associated mutant by substituting all the key hydrophobic residues to arginine (L90R, I91R, L94R, I101R, L104R, L121R and F122R). Unfortunately, both native and mutant scVps5₇₀₋₁₂₈ constructs expressed relatively poorly in bacteria, with

relatively low yields of purified proteins, so we decided to carry out this binding experiment using GST pull-down instead of ITC. Also, we used hRetromer in this experiment to resemble scRetromer. According to our result, the native GST- scVps5₇₀₋₁₂₈ construct containing the LF motif binds weakly to hRetromer compared to the control binding protein GST-hFAM21_{R19-R21}. While we observed some non-specific binding in the mutant and the GST control, the binding observed in the native construct suggests that the LF motif in scVps5 likely contributes to the high binding affinity to Retromer due to the enhanced avidity. We have now included the new binding data into the revised manuscript as Figure S10E, along with the following description in the result section.

“Using GST pull-down, we confirmed that hRetromer was able to bind to the GST-tagged scVps570-128 construct containing the LF motif (**Fig. S10E**). In comparison to the control GST-tagged hFAM21R19-R21 construct⁴³, the LF motif alone only binds weakly to VPS35, and we speculate that Vps5 orthologues containing these three sequences together will create a high binding affinity for Retromer due to enhanced avidity.”

Figure R2 (new Fig. S10E). GST pull-down showing the weak interaction between GST-tagged scVps5₇₀₋₁₂₆ LF motif containing loop and hRetromer. GST-hFAM21_{R19-R21} was used as the positive control.

2. The SMT assay shown in Figure 6, please incorporate peptides containing [K/R]Φ and LF motifs.

We appreciate the suggestion. However, we believe the use of the macrocyclic peptide RT-D3 (and the scrambled RT-D3 negative control) are sufficient in this instance to demonstrate the importance of the PL motif in recruiting scRetromer to the membrane.

Response to the authors: Same as 1. Experiments with site-specific mutations are absolutely needed to support the authors' main conclusion.

Again, we appreciate the concern. To address reviewer's concern, we applied the CRISPR-Cas9 system to generate yeast expressing Vps5 with N-terminal point mutations. After several attempts, we successfully obtained the quadruple mutant (L121G/F122G/F161E/L196R) of scVps5, which alters residues in the three main sites of predicted interaction. Additional SMT experiments using the quadruple mutant of Vps5 were carried out. The result is consistent with the addition of cyclic peptide RT-D3 data, showed a defect of coat formation. Addition control experiment was also performed to ensure that the quadruple mutant of scVps5 still forms heterodimer with scVps17. Overall, we believe that the three main sites of predicted interaction together are responsible for the pentameric complex formation. We have now included the new SMT data into the revised manuscript as **Figure S13A-S13D**, along with the following description in the result section.

"In the similar experiment where we applied the quadruple mutant Vps5 (L121G/F122G/F161E/L196R), which alters residues in the three main sites of predicted interaction - the PL motif, the [K/R] Φ motif, and the Vps35-binding LFa sequences, showed a defect of coat formation (**Figs. S13A and S13B**). The quadruple mutations on Vps5 do not impair heterodimer formation and able to form oligomer on the SMT (**Figs. S13C and S13D**)."

Figure R3 (new Fig. S13A – S13D). Effect of scVps5 mutant on scRetromer recruitment on supported membrane tubes. (A) Effect of the scVps5 mutant (L121G/F122G/F161E/L196R) on the recruitment of scRetromer-mClover to supported membrane tubes (SMTs). SMTs containing 5% PtdIns(3)P were formed and incubated with 25 nM scSNX-BAR and 25 nM scRetromer-mClover while being imaged by confocal microscopy. Frame rate: 0.2 Hz. Scale bar 2 μ m. Top scVps17/scVps5 wild-type, bottom mutant. See also **Movies S7 and S8**. **(B)** Quantification of the recruitment of scRetromer-GFP on the tube. Quantification was done using FIJI. Measuring the mean fluorescence signal on the tube over a 10 μ m length (N=13 (WT) and N=14 (mutant) tubes from 2 experiments). **(C)** SMTs were

incubated with 100 nM scSNX-BARs and imaged by confocal microscopy at a frame rate of 0.2 Hz. Red: Texas-Red DHPE. (D) Kymograph of SMT shown in (C).

Reviewer #2 (Remarks to the Author):

The authors addressed all my questions and added additional experimental details to the figures. I therefore have no further requests and recommend the study for publication.

Reviewer #3 (Remarks to the Author):

The authors have performed additional significant experiments and added comments to clarify and answer the different concerns on their manuscript. The presented work has thus gained relevance and is now fitted to be published in nature communications.

This will bring additional insights and knowledge to the field. I thereby thank the authors for improving their manuscript.

Reviewer #4 (Remarks to the Author):

I appreciate the authors' efforts in addressing my queries. The point that the rates of association appear different because the proteins are tagged with different fluorophores can easily be addressed by fitting the data to an exponential rise function and normalizing the data to the fitted plateau value. This would be a nice addition for the reader to evaluate if the rates of association are similar or different. Also, showing the control data set (in the absence of the R3D3 peptide) in the same plot would be useful for comparison. But these are just suggestions and I am otherwise satisfied with the responses provided.